# ASTRA: ACTIVATION-SPACE TAIL-EIGENVECTOR LOW-RANK ADAPTATION OF LARGE LANGUAGE MODELS

## ABSTRACT

Parameter-Efficient Fine-Tuning (PEFT) methods, especially LoRA, are widely used for adapting pre-trained models to downstream tasks due to their computational and storage efficiency. However, LoRA and its successors often focus on well-optimized principal subspaces of model activations, yielding diminishing returns and potentially destabilizing pretrained representations, while the subspaces correspond to tail eigenvectors remain largely under-utilized. In this work, we propose **Astra** (Activation-Space Tail-Eigenvector Low-Rank Adaptation), a novel PEFT method that leverages the tail eigenvectors of the model output activations—estimated from a small task-specific calibration set—to construct task-adaptive low-rank adapters. By constraining updates to the subspace spanned by the tail eigenvectors of output activations, Astra avoids interfering with pretrained task-relevant semantic structure and adapts in directions that minimize energy in the original task-specific representational space, leading to faster convergence and improved downstream performance. Extensive experiments across natural language understanding (NLU) and natural language generation (NLG) tasks demonstrate that Astra consistently outperforms existing PEFT baselines across 16 benchmarks and even surpasses full fine-tuning (FFT) in certain scenarios.

## 1 INTRODUCTION

Large Language Models (LLMs) have achieved remarkable success across a wide range of tasks (Achiam et al., 2023; Dubey et al., 2024; Guo et al., 2025). A common strategy for adapting these pretrained models to downstream tasks is full fine-tuning (FFT), in which all model parameters are updated. However, the substantial computational and memory costs associated with this process severely limit its practicality in resource-constrained environments (Singh et al., 2024; Liu et al., 2024a). To overcome these limitations, parameter-efficient fine-tuning (PEFT) methods have emerged as a promising alternative. By introducing a small number of additional trainable components while keeping the pretrained parameters frozen (Liu et al., 2021; Li & Liang, 2021; Hu et al., 2023b), PEFT methods significantly reduce the number of trainable parameters while maintaining competitive performance on downstream tasks.

Among various PEFT methods, LoRA (Hu et al., 2022) has gained considerable attention for its simplicity and effectiveness. A theoretical motivation for the effectiveness of LoRA is offered by the intrinsic dimension hypothesis (Li et al., 2018; Aghajanyan et al., 2021), which posits that the solution space of fine-tuning lies in a low-dimensional subspace. However, the default initialization scheme of LoRA often results in very small gradients in the early stages of training, potentially leading to slow convergence and suboptimal adaptation (Meng et al., 2024; Wang et al., 2024b)

To address these limitations, recent research has proposed alternative initialization strategies for LoRA, which can be broadly categorized into two types. **Weight-driven** approaches(Meng et al., 2024; Wang et al., 2024a) leverage the structure of pretrained weights to guide the initialization of low-rank adapters, whereas **data-driven** methods(Yang et al., 2024; Wang et al., 2024b; Paischer et al., 2024) utilize data distributions and task-specific signals for initialization. However, most existing works overlook two critical aspects: (1) The output activations of LLM exhibit low-rank structure, where the major components are captured in a low-dimensional subspace (Yu & Wu, 2023; Liu et al., 2024a). This principal low-rank subspace is progressively formed and optimized during pretraining to capture rich semantic information (Wu et al., 2024). However, further updates within this subspace

during fine-tuning yield diminishing returns, potentially disturbing the learned representations and causing unstable convergence (Kumar et al., 2022). (2) Meanwhile, the dimensions corresponding to tail eigenvalues remain under-utilized (Nayak et al., 2025). These observations suggest that adapting in such under-explored subspaces may increase the effective rank (Roy & Vetterli, 2007), enhancing task-specific representational capacity and improving model's adaptability to downstream tasks.

Building on this insight, we propose **Astra** (**A**ctivation-**S**pace **T**ail-Eigenvector Low-**R**ank **A**daptation), a novel PEFT method that exploits the under-explored tail subspaces of output activations to construct learnable adapters. Specifically, Astra begins by performing eigendecomposition on the covariance matrix of the output activations using a small task-specific calibration dataset $D$, i.e., $Cov(Y) = Q\Lambda Q^\top$, where $Q$ denotes the eigenvectors and $\Lambda$ is the diagonal matrix of corresponding eigenvalues. To constrain optimization within the under-explored subspaces, Astra then projects the weight matrix $W$ onto the subspace spanned by the tail eigenvectors, thereby deriving task-adaptive low-rank adapters aligned with the under-utilized activation directions, i.e., $A = Q_{[:,-r:]}^\top W$ and $B = Q_{[:,-r:]}$, where $r$ denotes the LoRA rank. This initialization strategy offers twofold advantages: (1) **Orthogonality to task-relevant pretrained semantic structure**: By initializing LoRA adapters in directions orthogonal to the principal activation subspace, Astra minimizes interference with the model's native task competence, ensuring stability and semantic consistency during fine-tuning. (2) **Energy-minimizing initialization**: Among all possible low-rank update directions, Astra selects those that minimize perturbation energy in the original task-relevant output space. This enables efficient adaptation by enhancing task-relevant representations along previously under-utilized dimensions, accelerating convergence and improving downstream performance.

We conduct extensive experiments on a wide range of tasks to evaluate the effectiveness of Astra, including natural language understanding (NLU) and multiple natural language generation (NLG) tasks such as *Mathematical Reasoning*, *Code Generation*, and *Commonsense Reasoning*. Experimental results demonstrate that Astra consistently outperforms existing PEFT baselines across 16 benchmarks and even surpasses full fine-tuning (FFT) on certain tasks. Our contributions can be summarized as follows:

- We propose **Astra**, a novel initialization method for LoRA that leverages the under-utilized eigenspace of output activations for low-rank adaptation. Astra provably preserves task-relevant representations and minimizes perturbation energy in the task-specific output space, enabling efficient and stable fine-tuning.

- We conduct extensive experiments on a wide range of NLU and NLG tasks, including general language understanding, mathematical reasoning, code generation, and commonsense reasoning. Extensive experimental results demonstrate that Astra consistently outperforms existing PEFT methods, demonstrating its effectiveness and adaptability.

- We present systematic ablations on eigenvectors, LoRA ranks, and calibration data, which consistently confirm the effectiveness and efficiency of our approach. In addition, effective rank analysis supports the core hypothesis that Astra enhances task-specific representational capacity while preserving pretrained semantics.

## 2 METHOD

### 2.1 PRELIMINARIES OF LoRA'S INITIALIZATION

LoRA (Hu et al., 2022) introduces trainable updates by reparameterizing weight modifications as the product of two low-rank matrices. Formally, given a pre-trained weight matrix $W_0 \in \mathbb{R}^{m \times n}$, LoRA decomposes the weight changes as:

$$\tilde{W} = W_0 + \Delta W = W_0 + \frac{\alpha}{r} BA \tag{1}$$

where $\Delta W$ denotes the weight change, which is decomposed into two low-rank matrices $B \in \mathbb{R}^{m \times r}$ and $A \in \mathbb{R}^{r \times n}$ with an intrinsic rank $r \ll \min(m, n)$, $\alpha$ is a scaling constant. This parameterization reduces the number of trainable parameters from $mn$ to $(m+n)r$, significantly improving fine-tuning efficiency. In practice, $A$ is initialized from the Gaussian distribution, while $B$ is initialized as an all-zero matrix to ensure that the initial model output remains unchanged. However, such random initialization can lead to slower convergence, as the gradients of the trainable adapters can be very small or in random directions during the early stages of fines-tuning (Meng et al., 2024).

## 2.2 ACTIVATION-SPACE TAIL-EIGENVECTOR LOW-RANK ADAPTATION

To address these challenges, we propose Astra, a novel initialization method for LoRA designed to enhance adaptation efficiency and stability. The method consists of two main steps: (1) orthogonal decomposition of the activation space to preserve original semantic structure, and (2) projection of the weights onto the tail subspace for LoRA initialization. Below, we describe each step in detail.

**Step 1: Orthogonal Decomposition.** Astra begins by collecting the covariance matrix of output activations that are relevant to the downstream task. Specifically, we randomly sample a small set of data (e.g., 64 samples) from the training set to form the calibration dataset. These samples are then passed to the LLM for forward propagation. Denoting $Y \in \mathbb{R}^{d_{out} \times N}$ as the output activation of a linear layer, where $N$ is the number of calibration samples, we compute the covariance matrix as:

$$\text{Cov}(Y) = \mathbb{E}[YY^\top] - \mathbb{E}[Y]\,\mathbb{E}[Y]^\top \tag{2}$$

where $\mathbb{E}[\cdot]$ denotes the expectation operator. Since $\text{Cov}(Y)$ is positive semi-definite, it can be decomposed as $\text{Cov}(Y) = Q\Lambda Q^\top$, where $Q \in \mathbb{R}^{d_{out} \times d_{out}}$ is an orthogonal matrix, and each eigenvector $q_i$ represents an orthogonal direction in the output activation space, defining an independent axis that captures a distinct mode of variation. We then perform an orthogonal decomposition of output activations $Y$ based on the eigenvectors, given by:

$$Y = Q_{[:,:d_{out}-r]}Q_{[:,:d_{out}-r]}^\top Y + Q_{[:,-r:]}Q_{[:,-r:]}^\top Y \tag{3}$$

$$= Q_{[:,:d_{out}-r]}Q_{[:,:d_{out}-r]}^\top (Wx + b) + Q_{[:,-r:]}Q_{[:,-r:]}^\top (Wx + b) \tag{4}$$

$$= \underbrace{Q_{[:,:d_{out}-r]}Q_{[:,:d_{out}-r]}^\top Wx + b}_{\text{Frozen}} + \underbrace{Q_{[:,-r:]}Q_{[:,-r:]}^\top Wx}_{\text{Trainable}} \tag{5}$$

**Theorem 2.1** (Tail-Space Ensures Orthogonality to Pretrained Semantics). *Suppose output activation $Y$ exhibits a low-rank structure and its primary information is captured in a low-dimensional principal subspace. Then, the **target fine-tuning space** that minimizes the impact on the existing semantic structure, i.e., $Y_{main}$, represented by the principal subspace is the orthogonal space spanned by the tail eigenvectors, i.e., $Y_{tail}$, given by $Y_{main}^\top Y_{tail} = 0$.*

*Proof.* Let $\Sigma \in \mathbb{R}^{d_{out} \times d_{out}}$ be a symmetric positive semi-definite matrix with eigendecomposition $\Sigma = Q\Lambda Q^\top$, where $Q = [Q_{\text{main}} \,|\, Q_{\text{tail}}]$ partitions the eigenvectors into main-$(d_{out} - r)$ and tail-$r$ components. For any $Y \in \mathbb{R}^{d_{out} \times N}$, define the projections as:

$$Y_{\text{main}} := Q_{\text{main}}Q_{\text{main}}^\top Y, \quad Y_{\text{tail}} := Q_{\text{tail}}Q_{\text{tail}}^\top Y.$$

where $Y_{\text{main}}$ captures the primary information. Since $Q$ is orthogonal, the submatrices $Q_{\text{main}}$ and $Q_{\text{tail}}$ form the orthonormal bases for the output activation covariance matrix, i.e. $Q_{\text{main}}^\top Q_{\text{tail}} = 0$. We expand the inner product between the principal and residual projections:

$$Y_{\text{main}}^\top Y_{\text{tail}} = (Q_{\text{main}}Q_{\text{main}}^\top Y)^\top (Q_{\text{tail}}Q_{\text{tail}}^\top Y) = Y^\top Q_{\text{main}}Q_{\text{main}}^\top Q_{\text{tail}}Q_{\text{tail}}^\top Y.$$

Since $Q_{\text{main}}^\top Q_{\text{tail}} = 0$, we have $Q_{\text{main}}Q_{\text{main}}^\top Q_{\text{tail}} = 0$, which implies:

$$Y_{\text{main}}^\top Y_{\text{tail}} = 0.$$

Thus, building on the projection decomposition in Eq.5, Astra is enforced to operate entirely within the residual activation subspace orthogonal to the dominant directions, thereby avoiding interference with existing semantic structure. □

**Step 2: Tail-Subspace Projection.** Then based on Eq.5, the initialization scheme of the two learnable low-rank matrices $A$ and $B$ in LoRA can be formally expressed as:

$$A_{\text{init}} = Q_{[:,-r:]}^\top W \in \mathbb{R}^{r \times d_{in}}. \tag{6}$$

$$B_{\text{init}} = Q_{[:,-r:]} \in \mathbb{R}^{d_{out} \times r}, \tag{7}$$

where $A_{\text{init}}$ and $B_{\text{init}}$ serve as the two learnable low-rank matrices in LoRA. Since the additional term $\Delta W = BA$ is non-zero at initialization, we adjust the frozen component to ensure that the original model outputs remain unchanged. Formally, this yields:

$$W' = W^{(0)} + \Delta W = \underbrace{(W^{(0)} - B_{\text{init}} A_{\text{init}})}_{\text{Frozen}} + \underbrace{B'A'}_{\text{Trainable}} \tag{8}$$

where the learnable matrices $A'$ and $B'$ parameterize the task-specific update $\Delta W$. By constraining optimization within the subspace spanned by the tail eigenvectors, Astra utilize the under-explored subspace and thus effectively improves adaptation efficiency and stability. Below, we present the detailed Algorithm 1, and a PyTorch-like implementation for Astra in provided in Appendix I.

---

**Algorithm 1:** Astra: Activation-Space Tail-Eigenvector Low-Rank Adaptation

---

**Input:** Model $M$, LoRA rank $r$, calibration data $x$, weight matrices $W \in \mathbb{R}^{d_{\text{out}} \times d_{\text{in}}}$
**Output:** Initialized parameters $W_{\text{frozen}}, A_{\text{init}}, B_{\text{init}}$
1: $\hat{Y} \leftarrow M(x; W)$              ▷ Forward propagation

2: $\text{Cov}(Y) \leftarrow \mathbb{E}[YY^{\top}] - \mathbb{E}[Y]\mathbb{E}[Y]^{\top}$

3: $\text{Cov}(Y) = Q\Lambda Q^{\top}$              ▷ Eigen-decomposition

4: Initialize trainable low-rank matrices:
   $A_{\text{init}} = Q_{[:,-r:]}^{\top} W \in \mathbb{R}^{r \times d_{\text{in}}}$
   $B_{\text{init}} = Q_{[:,-r:]} \in \mathbb{R}^{d_{\text{out}} \times r}$              ▷ Astra Initialization

5: Compute frozen and update terms:
   $W_{\text{frozen}} = W^{(0)} - B_{\text{init}} A_{\text{init}}$
   $W_{\text{trainable}} = B_{\text{init}} A_{\text{init}}$

**return** $W_{frozen}, A_{init}, B_{init}$

---

### 2.3 THEORETICAL ANALYSIS OF ASTRA INITIALIZATION

Below, we provide further theoretical guarantees demonstrating that the initialization schemes in Eq.6 and Eq.7 minimize perturbation energy with respect to the spectral geometry of the output activations, thereby preventing the disruption of the existing semantic structure and ensuring stable and efficient task adaptation.

**Theorem 2.2** (Tail-Space Minimizes Output Perturbation Energy). *Let $W \in \mathbb{R}^{d_{out} \times d_{in}}$ denote the pretrained model weights, and let $\Sigma_X := \mathbb{E}[XX^{\top}]$ be the input covariance matrix. The output activation can be defined covariance as:*

$$\Sigma_Y := \mathbb{E}[YY^{\top}] = W\Sigma_X W^{\top}.$$

*For any orthogonal projection $P = UU^{\top}$ with $U \in \mathbb{R}^{d_{out} \times r}$, $U^{\top}U = I$, define the low-rank initialization path as $\Delta W := UU^{\top}W$. The expected squared output perturbation is:*

$$\mathcal{E}(U) := \mathbb{E}_x[\|\Delta W x\|^2] = \text{Tr}(U^{\top}\Sigma_Y U).$$

*Then this energy is minimized when $U$ spans the eigenspace corresponding to the $r$ smallest eigenvalues of $\Sigma_Y$. That is:*

$$\text{Tr}(U^{\top}\Sigma_Y U) \geq \text{Tr}(Q_{tail}^{\top}\Sigma_Y Q_{tail}) = \sum_{i=d-r+1}^{d} \lambda_i,$$

*where $\lambda_1 \geq \cdots \geq \lambda_d$ are the eigenvalues of $\Sigma_Y$.*

The detailed proof of Theorem 2.2 is provided in Appendix B.

## 3 EXPERIMENTS

In this section, we provide a comprehensive evaluation of Astra from three perspectives. 1) We first assess the Natural Language Understanding (NLU) capabilities using the GLUE (Wang et al., 2018)

benchmark (Section 3.2). 2) Next, we evaluate the performance of our method on Natural Language Generation (NLG) tasks, covering mathematical reasoning, code generation, and commonsense reasoning (Section 3.3). 3) Finally, we conduct ablation studies to analyze the effectiveness of our approach with respect to varying eigenvectors, LoRA ranks and calibration datasets (Section 3.4). All experiments are conducted on NVIDIA A100-SXM4 (80GB) GPUs.

## 3.1 BASELINES

To substantiate the effectiveness of our method, we compare Astra against full fine-tuning (FFT), vanilla LoRA, and 6 representative LoRA variants. These variants can be grouped as follows:

1. **Weight-driven initialization variants**:
   - *PiSSA* (Meng et al., 2024) initializes adapters with principal components and freezes the residual.
   - *MiLoRA* (Wang et al., 2024a) initializes adapters with the smallest singular components.

2. **Data-driven initialization variants**:
   - *CorDA* (Yang et al., 2024) builds adapters conditioned on context for task-specific adaptations.
   - *LoRA-GA* (Wang et al., 2024b) constructs low-rank matrices by approximating the gradient from the first step of full fine-tuning.

3. **Other LoRA variants** (with modified structure, hyperparameters, etc.):
   - *rsLoRA* (Kalajdzievski, 2023) introduces a square-root scaling factor to LoRA.
   - *DoRA* (Liu et al., 2024b) decomposes pretrained weights into magnitude and direction components, tuning the magnitude and direction matrix separately.

## 3.2 NATURAL LANGUAGE UNDERSTANDING

**Models and Datasets.** We fine-tune the T5-base model (Raffel et al., 2020) on a subset of tasks from the GLUE benchmark (Wang et al., 2018), including MNLI, QNLI, SST-2, CoLA and MRPC. The model is evaluated on the corresponding development sets, and accuracy is reported as the evaluation metric for all tasks. Additional details regarding the benchmarks are presented in Appendix E.1.

**Implementation Details.** We follow the experimental setup described in (Wang et al., 2024b) to ensure a fair comparison. Specifically, we convert the labels into tokens (e.g., "positive" or "negative") and use the prompt tuning to fine-tune the model for 1 epoch on each dataset. The normalized probabilities assigned to these tokens are then used for classification. Further experimental setup and implementation details can be found in Appendix F.1.

Table 1: Performance of T5-base fine-tuned with different adaptation methods on 5 datasets of the GLUE benchmark. We report accuracy for all tasks, and the results are averaged over three runs with different random seeds. Bold values indicate the best performance.

| | #Params | MNLI 393k | SST-2 67k | QNLI 105k | CoLA 8.5k | MRPC 3.7K | Average |
|---|---|---|---|---|---|---|---|
| Full FT | 223M | $86.95_{\pm 0.04}$ | $97.02_{\pm 0.03}$ | $98.78_{\pm 0.02}$ | $84.52_{\pm 0.01}$ | $84.19_{\pm 0.05}$ | 90.29 |
| LoRA | 3.2M | $86.97_{\pm 0.01}$ | $96.62_{\pm 0.02}$ | $98.75_{\pm 0.03}$ | $49.95_{\pm 1.33}$ | $47.67_{\pm 0.06}$ | 75.99 |
| DoRA | 3.4M | $87.05_{\pm 0.02}$ | $\mathbf{97.19}_{\pm 0.01}$ | $98.79_{\pm 0.02}$ | $84.23_{\pm 0.03}$ | $49.88_{\pm 0.05}$ | 83.43 |
| rsLoRA | 3.2M | $87.06_{\pm 0.01}$ | $97.13_{\pm 0.02}$ | $98.79_{\pm 0.02}$ | $83.89_{\pm 0.02}$ | $49.63_{\pm 0.04}$ | 83.30 |
| PiSSA | 3.2M | $87.01_{\pm 0.01}$ | $97.08_{\pm 0.01}$ | $98.82_{\pm 0.01}$ | $84.80_{\pm 0.01}$ | $82.84_{\pm 0.01}$ | 90.11 |
| CorDA | 3.2M | $\mathbf{87.11}_{\pm 0.03}$ | $\mathbf{97.19}_{\pm 0.02}$ | $98.81_{\pm 0.05}$ | $84.71_{\pm 0.22}$ | $69.12_{\pm 0.23}$ | 87.39 |
| LoRA-GA | 3.2M | $87.07_{\pm 0.01}$ | $97.13_{\pm 0.02}$ | $\mathbf{98.83}_{\pm 0.01}$ | $84.76_{\pm 0.11}$ | $84.19_{\pm 0.14}$ | 90.40 |
| Ours | 3.2M | $87.09_{\pm 0.01}$ | $96.45_{\pm 0.01}$ | $\mathbf{98.83}_{\pm 0.01}$ | $\mathbf{87.87}_{\pm 0.06}$ | $\mathbf{88.36}_{\pm 0.12}$ | $\mathbf{91.72}$ |

**Main Results.** Table 1 presents the performance of T5-base fine-tuned with different adaptation methods on five GLUE datasets. Our proposed approach consistently surpasses existing baselines, achieving the highest average accuracy across all tasks. The improvement is particularly pronounced on low-resource datasets such as MRPC and CoLA, where effective utilization of gradient information plays a critical role. These results suggest that our method can fully exploit the limited training signals, leading to stable and fast convergence even under data-scarce conditions.

Table 2: Comparison of full fine-tuning (Full FT) and several LoRA variants on 2 mathematical reasoning and 4 code generation benchmarks. The best PEFT results are highlighted in **bold**.

| Model | Method | #Params | GSM8K | Math | HumanEval | HumanEval+ | MBPP | MBPP+ | Average |
|---|---|---|---|---|---|---|---|---|---|
| LLaMA2-7B | Full FT | 6738M | 58.76 | 12.04 | 32.9 | 31.1 | 43.9 | 36.8 | 35.92 |
| | LoRA | 320M | 41.40 | 5.42 | 22.0 | 20.1 | 34.9 | 27.2 | 25.17 |
| | MiLoRA | 320M | 39.12 | 5.06 | 20.1 | 18.9 | 36.8 | 29.4 | 24.90 |
| | PiSSA | 320M | 51.63 | 7.36 | 23.2 | 20.1 | 36.7 | 29.5 | 28.08 |
| | CorDA | 320M | 52.99 | 8.08 | **25.0** | **23.2** | 36.2 | 29.6 | 29.18 |
| | Ours | 320M | **55.19** | **8.98** | **25.0** | **23.2** | **38.4** | **31.2** | **30.33** |
| LLaMA3-8B | Full FT | 8366M | 75.36 | 24.04 | 56.7 | 53.7 | 64.0 | 54.5 | 54.72 |
| | LoRA | 336M | 73.31 | 24.24 | 53.7 | 48.8 | 65.6 | 54.8 | 53.41 |
| | MiLoRA | 336M | 73.24 | 23.90 | 52.4 | 48.2 | 68.3 | 56.1 | 53.69 |
| | PiSSA | 336M | 76.50 | 26.92 | 57.1 | 52.0 | 68.0 | 56.3 | 56.14 |
| | CorDA | 336M | 77.26 | 26.52 | 55.5 | 50.0 | 67.7 | 57.7 | 55.78 |
| | Ours | 336M | **77.56** | **27.92** | **57.7** | **53.0** | **68.4** | **58.2** | **57.13** |

Table 3: Zero-shot performance of LLaMA2-7B and LLaMA3-8B fine-tuned with different adaptation methods on seven commonsense reasoning benchmarks. The best PEFT results are shown in **bold**.

| Model | Method | #Params | BoolQ | PIQA | HellaSwag | WinoGrande | ARC-e | ARC-c | OBQA | Average |
|---|---|---|---|---|---|---|---|---|---|---|
| LLaMA2-7B | Full FT | 6738M | 82.81 | 75.08 | 55.57 | 73.64 | 72.69 | 41.72 | 32.00 | 61.93 |
| | LoRA | 320M | 79.97 | 78.35 | 57.30 | 68.82 | 78.07 | 46.16 | 32.40 | 63.01 |
| | PiSSA | 320M | 83.03 | 78.18 | 57.52 | 70.72 | 78.41 | 47.35 | 33.40 | 64.09 |
| | MiLoRA | 320M | 79.66 | 78.13 | **57.53** | 69.22 | 77.31 | 45.39 | 32.40 | 62.81 |
| | CorDA | 320M | 82.87 | 78.45 | 56.24 | **71.82** | 75.67 | 43.09 | 33.00 | 63.02 |
| | Ours | 320M | **83.76** | **78.51** | 57.28 | 71.74 | **79.63** | **48.72** | **34.20** | **64.83** |
| LLaMA3-8B | Full FT | 8366M | 82.14 | 68.82 | 49.30 | 66.06 | 65.95 | 38.05 | 31.60 | 57.42 |
| | LoRA | 336M | 85.02 | 79.76 | 59.88 | 74.74 | 82.53 | 53.50 | 34.00 | 67.06 |
| | PiSSA | 336M | **86.76** | 80.47 | **60.63** | 76.64 | 81.94 | 52.82 | 36.00 | 67.89 |
| | MiLoRA | 336M | 84.07 | 79.92 | 60.31 | 74.59 | 81.27 | 51.62 | 34.80 | 66.65 |
| | CorDA | 336M | 85.84 | **80.74** | 60.43 | 76.56 | 82.70 | **54.44** | 35.20 | 67.99 |
| | Ours | 336M | 86.48 | 80.41 | 60.02 | **78.22** | **82.87** | 53.99 | **36.60** | **68.37** |

## 3.3 NATURAL LANGUAGE GENERATION

**Models and Datasets.** We conduct experiments using LLaMA2-7B (Touvron et al., 2023) and LLaMA3-8B (Dubey et al., 2024) across three NLG tasks: **Math**, **Code** and **Commonsense**.

- *Math:* For mathematical reasoning tasks, the models are fine-tuned on the MetaMathQA dataset (Yu et al., 2023) and evaluated on two widely used benchmarks, GSM8K (Cobbe et al., 2021) and MATH (Hendrycks et al., 2021), using PASS@1 accuracy as the evaluation metric.

- *Code:* To evaluate programming proficiency, we fine-tune the models on the CodeFeedback-Python105k dataset (Zheng et al., 2024) and assess performance on HumanEval (Chen et al., 2021) and MBPP (Austin et al., 2021) benchmarks. Additionally, we employ the EvalPlus framework (Liu et al., 2023) to test on the extended versions of these datasets, namely MBPP+ and HumanEval+, which provide a greater number of test cases compared to the original versions. We report the PASS@1 metric for these evaluations.

- *Commonsense:* For commonsense reasoning, the models are fine-tuned on the Commonsense170K dataset (Hu et al., 2023a) and tested on seven established benchmarks—BoolQ (Clark et al., 2019), PIQA (Bisk et al., 2020), HellaSwag (Zellers et al., 2019), WinoGrande (Sakaguchi et al., 2020), ARC-e, ARC-c (Clark et al., 2018), and OpenBookQA (Mihaylov et al., 2018). All tasks are tested in a zero-shot setting using the LM-Evaluation-Harness framework (Gao et al., 2024).

**Implementation Details.** To ensure a fair comparison, we adopt the experimental configurations delineated in (Meng et al., 2024; Wang et al., 2024b; Yang et al., 2024). Specifically, we set the LoRA rank to 128, with the LoRA alpha consistently equal to the rank, and insert adapters into all linear layers of the base model. All the experiments were conducted on the first 100,000 samples from each dataset and trained for one epoch to reduce computational overhead. Additional implementation details are provided in the Appendix F.2.

**Main Results.** Table 2 summarizes the results on mathematical reasoning and code generation tasks, and Table 3 reports the performance on commonsense reasoning benchmarks. Overall, our

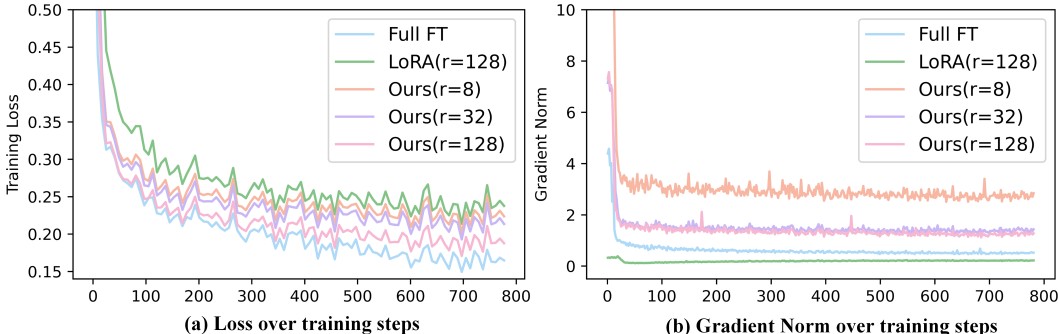

Figure 1: Training loss and gradient norm curves for FFT, LoRA (rank=128), and Astra with varying ranks on the MetaMathQA dataset. Our method (rank=8) performs even better+ than LoRA (rank=128), and higher ranks lead to faster loss reduction, approaching the performance of FFT.

approach consistently surpasses existing PEFT baselines, demonstrating robust generalization across diverse task categories. Below, we provide a breakdown of the results by task type:

- *Math:* Astra outperforms all other PEFT baselines on both the GSM8K and MATH datasets, achieving the best results overall, with the exception of a slight gap compared to Full FT on LLaMA2-7B. Figure 1 illustrates the loss curves and gradient norm trends during fine-tuning of LLaMA2-7B on the MetaMathQA dataset. Notably, Astra (with rank=8) converges faster than LoRA (rank=128), highlighting its efficiency in downstream task adaptation with minimal resources.

- *Code:* For code generation tasks, Astra also achieves outstanding results, even surpassing Full FT on LLaMA3-8B. Our method shows remarkable programming proficiency, as reflected in the results across HumanEval and MBPP benchmarks.

- *Commonsense:* Astra demonstrates consistently strong performance across seven commonsense reasoning benchmarks. Although it slightly lags on HellaSwag, it achieves the best overall average performance among all baselines.

### 3.4    ABLATION STUDIES

**Eigenvectors.**    To investigate the impact of eigenvectors corresponding to eigenvalues of varying magnitudes on fine-tuning performance, we initialize the adapters injected into LLaMA2-7B with eigenvectors selected from different quantiles of the eigenvalue spectrum. Specifically, we use eigenvectors corresponding to the top, tail, middle, lower quartile, and upper quartile eigenvalues, as well as randomly selected eigenvectors. The models are then fine-tuned on the MetaMathQA dataset and evaluated on the GSM8K and MATH benchmarks. As shown in Table 4, adapters initialized with tail eigenvectors achieve the best performance on both benchmarks, underscoring the efficacy of our strategy in leveraging tail eigenvectors from activation-space for fine-tuning.

Table 4: Performance of LLaMA2-7B fine-tuned with adapters initialized using eigenvectors from different quantiles of the eigenvalue spectrum.

| Eigenvectors | GSM8K | MATH |
|---|---|---|
| Random | 40.49 | 5.64 |
| Top | 40.71 | 5.48 |
| Upper Quartile (Q3) | 40.49 | 5.64 |
| Medium | 38.74 | 5.60 |
| Lower Quartile (Q1) | 42.76 | 5.70 |
| Tail | **55.19** | **8.98** |

**LoRA Rank.**    In this experiment, we explore the effects of varying LoRA rank from 8 to 128, aiming to assess whether our approach consistently outperforms other PEFT baselines across different rank values. Following the setup described in Section 3.3, we fine-tune LLaMA2-7B on the MetaMathQA dataset and evaluate it on the GSM8K and MATH benchmarks. Figures 2 (a)-(b) show that Astra consistently outperforms alternative PEFT methods with the same number of trainable parameters. Figure 2 (c) illustrates the final training loss across different ranks, demonstrating that our method achieves a better fit to the training data compared to LoRA, PiSSA, and CorDA. It is noteworthy that our approach outperforms LoRA at rank = 128 even with rank = 8, underscoring its efficiency in achieving better performance with fewer trainable parameters.

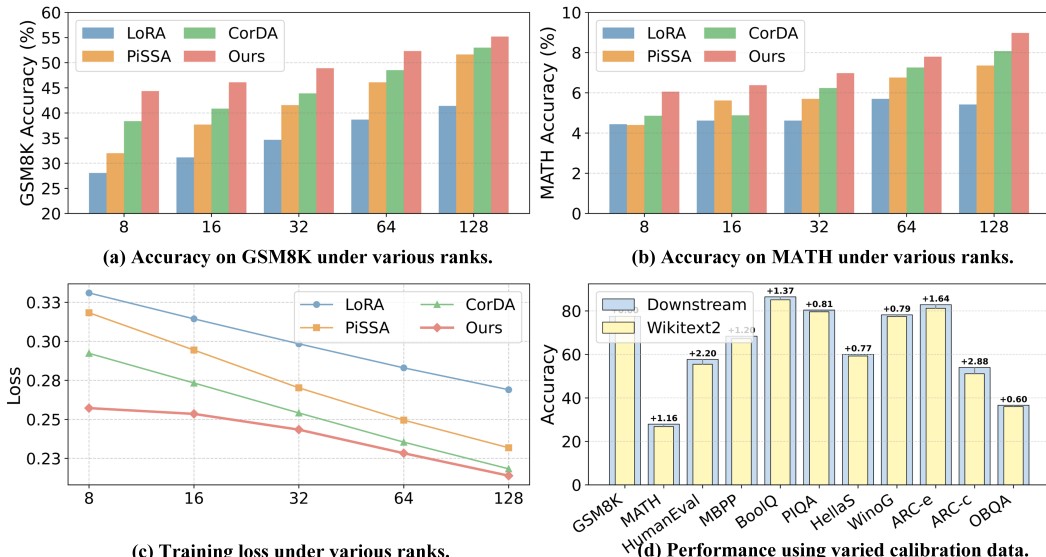

(a) Accuracy on GSM8K under various ranks.

(b) Accuracy on MATH under various ranks.

(c) Training loss under various ranks.

(d) Performance using varied calibration data.

Figure 2: (a) and (b) report the performance of different LoRA variants on GSM8K and MATH under various ranks, respectively. (c) shows the final training loss on the MetaMathQA dataset under various ranks. (d) illustrates the performance using different calibration data.

**Calibration Data.** To assess the robustness of Astra with respect to the calibration datasets, we conduct experiments using a general-purpose dataset (i.e. Wikitext-2) for calibration, and compare it with the default setting, where the downstream training set itself is used for calibration. The results, presented in Figure 2 (d), demonstrate that Astra achieves stable performance across different calibration datasets, while leveraging the downstream training set yields marginally better results.

## 4 DISCUSSION

**Enhancing Representation Capacity via Increased Effective Rank.** To evaluate the improvement in representational capacity introduced by our approach, we employ *effective rank*(Roy & Vetterli, 2007) as a metric to characterize the spectral structure of output activations before and after fine-tuning. Formally, the effective rank is defined as:

$$\mathcal{R}_{X,i} = \exp\left(-\sum_{j=1}^{d_{out}} \tilde{\lambda}_j \ln(\tilde{\lambda}_j)\right) \quad \text{and} \quad \tilde{\lambda}_j = \frac{\lambda_j}{\sum_{k=1}^{d_{out}} \lambda_k} \tag{9}$$

where $\lambda_j$ denotes the eigenvalues obtained from the eigendecomposition of the output activation covariance matrix, $X \in \{Q, K, V, O, Up, Down\}$ represents the projection layer type within the Transformer architecture, and $i$ indexes the corresponding Transformer layer.

A higher effective rank indicates that the output activations are distributed across more directions in the feature space, suggesting a richer and more diverse representational capacity. Conversely, a lower effective rank (only a few eigenvalues are large) implies that the activations are concentrated along a few dominant directions, reflecting more constrained or redundant representations (Li et al., 2025).

For each layer type, we aggregate the effective rank across all layers and compute the total, which is then compared before and after fine-tuning to quantify the overall change. As shown in Figure 3, both LoRA and Astra lead to an increase in effective rank. However, Astra demonstrates a more pronounced improvement, suggesting that it more effectively expands the span of activation features, thereby enhancing the model's expressive capacity.

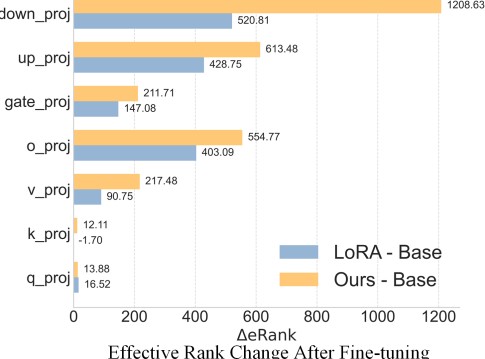

Figure 3: Comparison of effective rank before and after fine-tuning.

**Bridging PEFT and ReFT.** Representation Fine-Tuning (ReFT; Wu et al., 2024) has recently been proposed as a paradigm that departs from traditional parameter-efficient fine-tuning (PEFT) methods. Instead of directly modifying model parameters, ReFT operates in the representation space by applying lightweight transformations to intermediate activations. This design preserves the representational information acquired during pretraining and enables stable adaptation even in data-scarce scenarios. The connection between Astra and ReFT is shown in the following formulations:

$$Y = \underbrace{Q_{[:,:d_{\text{out}}-r]}Q_{[:,:d_{\text{out}}-r]}^{\top}Y}_{Y_{\text{main}}} + \underbrace{Q_{[:,-r:]}Q_{[:,-r:]}^{\top}Y}_{Y_{\text{tail}}} \qquad \triangleright \text{Astra}$$

$$= \underbrace{Y}_{Frozen} + \underbrace{R^{\top}(WY + b - RY)}_{Trainable} \qquad \triangleright \text{ReFT}$$

where $Y_{\text{main}}$ encodes major semantic information, i.e., $Y_{\text{main}} \approx Y$ and $R \in \mathbb{R}^{r \times d}$ is a learnable projection matrix with an orthogonality constraint, ensuring that updates remain separated from the dominant activation directions and thus avoid interfering with the pretrained semantic structure. When $R = W = Q[:, -r :]^{\top}$ and $b = 0$, the formulation is identical to Astra's initialization form. Astra therefore can be viewed as a method that inherits the representational advantages of ReFT while simultaneously retaining the flexibility of PEFT. Specifically, Astra enforces low-rank adaptation entirely within the tail activation subspace, which are orthogonal to the dominant directions of pretrained representations. This design introduces minimum perturbation energy and avoids interference with the pretrained semantics. Further discussion is provided in Appendix C.

## 5 RELATED WORK

**PEFT.** Parameter-efficient fine-tuning (PEFT) offers a lightweight alternative to full fine-tuning by updating only a small subset of parameters, effectively reducing computational overhead while maintaining strong performance in downstream task adaptation. PEFT methods can be broadly categorized into prompt-based (Lester et al., 2021; Li & Liang, 2021; Liu et al., 2021), adapter-based (Houlsby et al., 2019; Rücklé et al., 2020; Hu et al., 2023b), and LoRA-based approaches (Hu et al., 2022; Dettmers et al., 2023). Prompt-based methods introduce trainable tokens or embeddings that are prepended to the input or intermediate representations, while Adapter-based methods insert small trainable modules within each transformer layer to adapt the pre-trained model to new tasks.

**LoRA and Its Variants.** Among these methods, low-rank adaptation (LoRA) has gained particular attention for enabling effectively fine-tuning without modifying the original architecture or introducing additional inference latency (Li et al., 2018; Aghajanyan et al., 2021). Building on the success of LoRA, subsequent research has explored a variety of directions to improve its effectiveness and flexibility. Several works have investigated ***dynamic rank allocation*** (Valipour et al., 2022; Zhang et al., 2023b; Liu et al., 2024c), aiming to better balance expressivity and parameter efficiency. For instance, AdaLoRA (Zhang et al., 2023b) adaptively allocates parameter budgets across weight matrices based on their importance scores. In addition, ***structural modifications*** of LoRA (Liu et al., 2024b; Feng et al., 2024; Li et al., 2024) have been proposed to generalize LoRA beyond its original design. For example, DoRA (Liu et al., 2024b) decouples the learning process into magnitude and direction. Some research also focuses on optimizing the ***hyperparameters*** within LoRA to enhance fine-tuning efficiency and stability (Kalajdzievski, 2023; Hayou et al., 2024). For example, LoRA+ (Hayou et al., 2024) introduces differential learning rates for the low-rank matrices A and B, with a higher learning rate for B to accelerate convergence. Another line of work focuses on improving ***initialization strategies*** to stabilize training and accelerate convergence (Meng et al., 2024; Wang et al., 2024a; Yang et al., 2024). For instance, PiSSA (Meng et al., 2024) and LoRA-GA (Wang et al., 2024b) conduct SVD on pretrained weights and sampled gradients to initialize the low rank adapters of LoRA. We provide a detailed comparison between these LoRA variants in Appendix D.

## 6 CONCLUSION

In this paper, we proposed Astra, a novel PEFT method that leverages the under-explored tail eigenspace of output activations for low-rank adaptation. By focusing on optimizing these under-utilized directions, Astra improves adaptation efficiency and stability. Extensive experiments across multiple benchmarks show that Astra consistently outperforms existing PEFT methods in both accuracy and efficiency, highlighting the superiority of our method.

ETHICS STATEMENT

Our research adheres to the Code of Ethics, prioritizing transparency, responsible data usage, and careful consideration of potential social impacts.

All datasets employed in this work are publicly available and have been properly cited, ensuring full compliance with data usage agreements and privacy regulations. We acknowledge the critical importance of ethical AI development and are committed to aligning our work with responsible practices. The proposed Astra method, by optimizing pretrained LLMs within the under-utilized activation spaces, enhances the model's adaptability and representational capacity while preserving the integrity of the underlying pretrained task-relevant semantic structure. However, we remain cognizant of the potential for unintended consequences and emphasize the need for continued reflection on the broader social implications of such advancements in AI.

REPRODUCIBILITY STATEMENT

To ensure the reproducibility of our work, we provide all necessary resources to facilitate the replication of our results. The pseudocode 1 is presented in Section 2, and a PyTorch-like implementation of the Astra algorithm is provided in Appendix I. Additionally, we provide an anonymous repository of our code, which can be accessed at https://anonymous.4open.science/r/Anonymous-Astra/.

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

# The Supplementary Material for The Paper
# "Astra: Activation-Space Tail-Eigenvector Low-Rank Adaptation for Large Language Models"

- In Section A, we clarify the role of large language models (LLMs) in this work, explicitly stating that they were used only for language polishing and grammar refinement.

- In Section B, we provide detailed proof of Theorem 2.2 to better understand the rationale behind Astra's initialization strategy.

- In Section C, we provide a detailed discussion of how Astra represents a unique intersection between parameter-efficient fine-tuning (PEFT) and representation fine-tuning (ReFT)

- In Section D, we provide a systematic comparison of existing LoRA variants to highlight their relative strengths and differences.

- In Section E, we provide detailed descriptions of the benchmark datasets used in our evaluation. These descriptions cover the domains, sizes, and task characteristics of the datasets employed in both NLU and NLG experiments.

- In Section F, we present the implementation details in the main text. This includes the hyperparameter configurations and training details in our experiments.

- In Section G, we present additional experimental results that complement the findings reported in the main text. These include loss curves, more detailed evaluation outcomes, and further analyses of our method's behavior.

- In Section H, we present a series of case studies to demonstrate the improved performance in instruction-following of models that have been fine-tuned with Astra, providing qualitative evidence that complements the quantitative results in the main text.

- In Section I, we present a PyTorch-like implementation sketch to clarify the workflow, which facilitates reproducibility and bridge the gap between the theoretical formulation introduced in the main text and its practical implementation.

## A  THE USE OF LARGE LANGUAGE MODELS (LLMs)

We disclose the use of large language models (LLMs) in preparing this manuscript. They were employed solely for language polishing and grammar refinement, while all scientific content, ideas, and analyses were conceived and performed solely by the authors.

## B  PROOF OF THEOREMS

**Theorem 2.2.** *Let $W \in \mathbb{R}^{d_{out} \times d_{in}}$ denote the pretrained model weights, and let $\Sigma_X := \mathbb{E}[XX^\top]$ be the input covariance matrix. The output activation can be defined covariance as:*

$$\Sigma_Y := \mathbb{E}[YY^\top] = W\Sigma_X W^\top.$$

*For any orthogonal projection $P = UU^\top$ with $U \in \mathbb{R}^{d_{out} \times r}$, $U^\top U = I$, define the low-rank initialization path as $\Delta W := UU^\top W$. The expected squared output perturbation is:*

$$\mathcal{E}(U) := \mathbb{E}_x[\|\Delta W x\|^2] = \mathrm{Tr}(U^\top \Sigma_Y U).$$

*Then this energy is minimized when $U$ spans the eigenspace corresponding to the $r$ smallest eigenvalues of $\Sigma_Y$. That is:*

$$\mathrm{Tr}(U^\top \Sigma_Y U) \geq \mathrm{Tr}(Q_{tail}^\top \Sigma_Y Q_{tail}) = \sum_{i=d-r+1}^{d} \lambda_i,$$

*where $\lambda_1 \geq \cdots \geq \lambda_d$ are the eigenvalues of $\Sigma_Y$, and $Q_{tail} := [q_{d-r+1}, \ldots, q_d]$ are the corresponding eigenvectors.*

*Proof.* We begin with the definition of the perturbation energy:

$$\mathcal{E}(U) := \mathbb{E}_x[\|\Delta W x\|^2], \quad \text{with} \quad \Delta W := UU^\top W.$$

By the linearity of expectation and standard matrix norm identities, we express the expected energy as:

$$\mathcal{E}(U) = \mathbb{E}_x[\|UU^\top W x\|^2] = \text{Tr}(UU^\top W \Sigma_X W^\top UU^\top).$$

We simplify this using the cyclic property of the trace:

$$\mathcal{E}(U) = \text{Tr}(U^\top W \Sigma_X W^\top U) = \text{Tr}(U^\top \Sigma_Y U),$$

where we define the output activation covariance as:

$$\Sigma_Y := W \Sigma_X W^\top.$$

Thus, minimizing the output perturbation energy reduces to the classical trace-form optimization:

$$\min_{U^\top U = I} \text{Tr}(U^\top \Sigma_Y U).$$

Since $\Sigma_Y$ is symmetric and positive semi-definite, this trace-form optimization falls under the classical Ky Fan minimum trace theorem. It states that for any $U \in \mathbb{R}^{d \times r}$ with $U^\top U = I$, the trace $\text{Tr}(U^\top \Sigma_Y U)$ is minimized when $U$ spans the eigenspace corresponding to the $r$ smallest eigenvalues of $\Sigma_Y$.

Formally, if $\Sigma_Y = Q\Lambda Q^\top$ is the eigendecomposition with eigenvalues $\lambda_1 \geq \cdots \geq \lambda_d$, and $Q_{\text{tail}} := [q_{d-r+1}, \ldots, q_d]$, then:

$$\text{Tr}(U^\top \Sigma_Y U) \geq \text{Tr}(Q_{\text{tail}}^\top \Sigma_Y Q_{\text{tail}}) = \sum_{i=d-r+1}^{d} \lambda_i,$$

with equality if and only if $U = Q_{\text{tail}}$. This completes the proof. $\square$

## C  BRIDGING REFT AND PEFT

Representation Fine-Tuning (ReFT; Wu et al., 2024) was recently proposed as a departure from classical PEFT methods by shifting adaptation from the parameter space into the representation space. Instead of directly modifying model weights, ReFT can be formalized as learning a lightweight transformation to the pretrained activations:

$$\Phi_{\text{ReFT}}(Y) = Y + \Delta(Y; \theta), \tag{10}$$

where $Y$ denotes hidden activations, and $\Delta(\cdot; \theta)$ is a lightweight trainable function applied to intermediate representations while keeping all pretrained parameters frozen. This formulation highlights the central idea of ReFT: task-specific adaptation is achieved entirely through controlled interventions on hidden states. Building on this paradigm, LoReFT (Low-rank Linear Subspace ReFT) introduces additional structure by restricting the intervention to a low-rank subspace. Specifically, LoReFT defines

$$\Phi_{\text{LoReFT}}(Y) = Y + R^\top(WY + b - RY), \tag{11}$$

where $R \in \mathbb{R}^{r \times d}$ has orthonormal rows, and $W \in \mathbb{R}^{r \times d}$ together with $b \in \mathbb{R}^r$ parameterize the projected source. The orthogonality constraint ensures that learned updates remain disentangled from dominant directions in the hidden space, thereby preserving pretrained semantics while still enabling effective adaptation:

$$Y = Y + R^\top(WY + b - RY) \qquad\qquad \triangleright \text{LoReFT}$$

$$= Y - R^\top RY + R^\top(WY + b) \qquad\qquad \triangleright \text{LoReFT}$$

$$= \underbrace{Q_{[:,:d_{\text{out}}-r]}Q_{[:,:d_{\text{out}}-r]}^\top Y}_{Y_{\text{main}}} + \underbrace{Q_{[:,-r:]}Q_{[:,-r:]}^\top Y}_{Y_{\text{tail}}} \qquad \triangleright \text{Astra}$$

where $Y_{\text{main}}$ encodes major semantic information, i.e., $Y_{\text{main}} \approx Y$. It is worth noting that when $R = W = Q[:,-r:]^\top$ and $b = 0$, the formulation of LoReFT is identical to the initialization form of Astra. This equivalence indicates that Astra inherits the advantages of ReFT, particularly in terms of modifying activation representations in a controlled manner. At the same time, Astra remains fundamentally a PEFT method, as it introduces low-rank updates only within the subspace spanned by the tail eigenvectors. This design enables Astra to leverage the strengths of both paradigms: the stability and semantic preservation of ReFT, and the parameter-efficient flexibility of PEFT.

## D    OVERVIEW AND COMPARISON OF LoRA VARIANTS

To highlight the effectiveness and robustness of our approach, we compare Astra against a diverse set of LoRA variants. Below, we classify the baseline methods discussed in this work according to the types of modifications they introduce to vanilla LoRA, grouping them into four main categories:

1. **Initialization**:
   – *PiSSA* (Meng et al., 2024) applies singular value decomposition (SVD) to extract the principal singular values and vectors of the original weights. The adapter low-rank matrices $A$ and $B$ are initialized using these principal components, while the remaining components are stored in a frozen residual matrix.
   – *MiLoRA* (Wang et al., 2024a) diverges from PiSSA by applying adaptation exclusively to the subspace associated with the smallest singular values and maintaining the principal ones unchanged.
   – *CorDA* (Yang et al., 2024) introduces context-oriented decomposition adaptation, which builds task-aware adapters by orienting weight decomposition with the covariance of input activations. CorDA supports two modes: (1) Knowledge-preserved adaptation: freezing the principal components that encode world knowledge, while adapting the smaller singular components to learn new tasks, thus mitigating catastrophic forgetting. (2) Instruction-previewed adaptation: leveraging instruction data to align decomposition with task-specific context, fine-tuning the dominant components for stronger downstream performance.
   – *LoRA-GA* (Wang et al., 2024b) aligns the gradients of the low-rank matrices with those of full fine-tuning from the very first step. Concretely, it computes the eigenvectors of the gradient matrix via SVD and uses them to initialize the adapter matrices $A$ and $B$, ensuring that the initial update of $BA$ closely matches the direction of $\Delta W$ in full fine-tuning.

2. **Structure**:
   – *DoRA* (Liu et al., 2024b) decomposes pretrained weights into magnitude and direction components, fine-tuning the magnitude vector and applying low-rank adaptation solely to the directional component to improve capacity.
   – *MixLoRA* (Li et al., 2024 fuses multiple LoRA-based experts with a shared feed-forward (FFN) layer of the pretrained dense model, making it closer in design to high-performance Mixture-of-Expert systems.

3. **Hyperparameters**:
   – *rsLoRA* (Kalajdzievski, 2023) revisits the scaling factor in LoRA and theoretically proves that the stable choice should instead be $\gamma_r = \frac{\alpha}{\sqrt{r}}$ ensuring that both forward activations and backward gradients remain rank-stabilized across different $r$ values.
   – *LoRA-FA* (Zhang et al., 2023a)introduces a memory-efficient variation of LoRA by selectively freezing one of the two low-rank projection matrices. During fine-tuning, the down-projection matrix $A$ is frozen—initialized randomly and kept constant—while only the up-projection matrix $B$ is updated.

4. **Rank Allocation**:
   – *AdaLoRA* (Zhang et al., 2023b) parameterizes updates via a pseudo-SVD $P\Lambda Q$ and adaptively prunes singular values based on importance scores to allocate the LoRA rank budget across layers according to task relevance.
   – *DyLoRA* (Valipour et al., 2022) introduces a dynamic, search-free extension of LoRA that eliminates the need for exhaustive rank tuning. Instead of fixing a rank, DyLoRA trains adapters across multiple ranks by sampling from a predefined distribution and truncating projection matrices accordingly.

Since our method also belongs to the initialization category, we present a detailed comparison of representative LoRA initialization variants in Table 5, highlighting their key design differences.

## E    DETAILS OF BENCHMARK DATASETS

In this section, we provide an overview of the benchmark datasets employed in our experiments.

Table 5: Comparison of our selective LoRA initialization variants in the experimental section.

| Method | Driven-Type | Signal | Gradient Free | Calibration Data |
|--------|-------------|--------|---------------|------------------|
| PiSSA | weight | weight | ✔ | No |
| MiLoRA | weight | weight | ✔ | No |
| CorDA | data | Input Context | ✔ | Downstream |
| LoRA-GA | data | Gradient | ✘ | Downstream |
| Astra | data | Output Activation | ✔ | Downstream/General |

## E.1 BENCHMARKS OF NATURAL LANGUAGE UNDERSTANDING

For NLU tasks, we use a subset of the GLUE benchmark (Wang et al., 2018) in our experiments, including CoLA, SST-2, MRPC, MNLI and QNLI. We present the statistical information of these datasets in Table 6 below.

Table 6: Statistical overview of the GLUE benchmark datasets used in our experiments.

| Corpus | Task | #Train | #Val | #Test | #Labels | Domain |
|--------|------|--------|------|-------|---------|--------|
| CoLA | Acceptability | 8.55k | 1.04k | 1.06k | 2 | misc. |
| SST-2 | Sentiment | 67.3k | 872 | 1.82k | 2 | Movie Reviews |
| MRPC | Paraphrase | 3.67k | 408 | 1.73k | 2 | News |
| MNLI | NLI | 393k | 19.65k | 19.65k | 3 | misc. |
| QNLI | QA/NLI | 105k | 5.46k | 5.46k | 2 | Wikipedia |

## E.2 BENCHMARKS OF NATURAL LANGUAGE GENERATION

For NLG tasks, we evaluate models across three key dimensions—Mathematical Reasoning, Code Generation, and Commonsense Reasoning—using the following benchmark datasets:

1. **Mathematical Reasoning**:

   – *MetaMathQA* (Yu et al., 2023) is a large-scale dataset (395k) derived via augmentation of GSM8K and MATH training sets, designed to enhance mathematical reasoning capabilities

   – *GSM8K* (Cobbe et al., 2021) is a rigorously curated dataset of approximately 8.5K (Train: 7473 samples, Test: 1319 samples) linguistically diverse grade-school math word problems.

   – *MATH* (Hendrycks et al., 2021) is a challenging benchmark consisting of approximately 12,500 (Train: 7500 samples, Test: 5000 samples) contest-level mathematics problems, covering topics ranging from algebra and geometry to number theory and pre-calculus.

2. **Code Generation**:

   – *CodeFeedback-Python105k* (Zheng et al., 2024) is a high-quality subset extracted from the CodeFeedback-Filtered-Instruction collection (Zheng et al., 2024) and curated for Python-based code generation tasks. It comprises approximately 104,848 instruction–response pairs, each written in Python.

   – *HumanEval* (Chen et al., 2021) is a benchmark of 164 Python programming problems, each requiring a function as the solution, which is widely adopted for evaluating functional correctness of code generated by language models.

   – *MBPP* (Austin et al., 2021) contains 974 short Python programming tasks designed for entry-level coders. Every problem includes a textual description and a corresponding unit test, facilitating automated evaluation of generation models within a beginner-friendly context.

3. **Commonsense Reasoning**:

   – *BoolQ* (Clark et al., 2019) is a yes/no question answering dataset containing naturally occurring queries, designed to assess a model's ability to handle open-ended binary classification.

- *PIQA* (Bisk et al., 2020) evaluates physical commonsense reasoning through multiple-choice questions, where each query is paired with two candidate answers requiring intuitive physical knowledge.
- *HellaSwag* (Zellers et al., 2019) focuses on commonsense inference, providing a context followed by several possible continuations, with the task being to select the most plausible ending.
- *WinoGrande* (Sakaguchi et al., 2020) introduces large-scale fill-in-the-blank questions with two options, targeting pronoun resolution and commonsense disambiguation.
- *ARC-e* and *ARC-c* (Clark et al., 2018) are the Easy and Challenge subsets of the ARC dataset, composed of grade-school science multiple-choice questions. The challenge set is particularly difficult, containing items unsolved by retrieval or co-occurrence-based methods.
- *OpenBookQA* (Mihaylov et al., 2018) comprises elementary-level science questions requiring multi-step reasoning. Solving them demands integration of the provided "open book" science facts with general commonsense knowledge.

## F EXPERIMENTAL SETUP AND IMPLEMENTATION DETAILS

To ensure a fair comparison, all experimental setups are consistent across all methods. In the following, we describe the experimental setup and hyperparameters configuration in detail.

### F.1 EXPERIMENTAL DETAILS OF NLU

For natural language understanding (NLU) tasks, we apply low-rank adaptation to all the linear modules in T5-base except for the embedding layer and language model head. For FFT, LoRA, and its variants, we use a learning rate of $1 \times 10^{-4}$, while for DoRA (Liu et al., 2024b), a learning rate of $2 \times 10^{-4}$ is employed to adhere to the settings in the original paper. The LoRA rank is set to 8, and the LoRA $\alpha$ is set to 16. The detailed configurations are depicted in Table 7.

### F.2 EXPERIMENTAL DETAILS OF NLG

For natural language generation (NLG) tasks, we utilize the AdamW (Loshchilov & Hutter, 2017) optimizer with a batch size of 128 and a learning rate of 2e-5. A cosine annealing schedule with a warmup ratio of 0.03 is applied without incorporating weight decay. To reduce computational overhead, model parameters are stored in `bfloat16` precision. The LoRA alpha $\alpha$ is set consistently equal to the LoRA rank $r$. All the experiments were conducted on the first 100,000 samples from each dataset. Table 8 summarizes the detailed configurations.

Table 7: Experimental setup and hyperparameters configurations for NLU tasks

| hyperparameters | setup |
|---|---|
| batch size | 128 |
| epochs | 1 |
| learning rate | 1e-04
DoRA: 2e-4 |
| max length | 128 |
| lr scheduler | cosine |
| warmup ratio | 0.03 |
| weight decay | 0.00 |
| data type | float32 |
| LoRA rank | 8 |
| LoRA alpha | 16 |
| LoRA dropout | 0.00 |
| target modules | q, k, v, o,
wi_0, wi_1, wo |

Table 8: Experimental setup and hyperparameters configurations for NLG tasks

| hyperparameters | setup |
|---|---|
| batch size | 128 |
| epochs | 1 |
| learning rate | 2e-05 |
| max sequence length | 512 |
| lr scheduler | cosine |
| warmup ratio | 0.03 |
| weight decay | 0.00 |
| data type | bfloat16 |
| LoRA rank | 128 |
| LoRA alpha | 128 |
| LoRA dropout | 0.00 |
| target modules | q_proj, k_proj, v_proj,
o_proj, gate_proj,
up_proj, down_proj |

## G  ADDITIONAL EXPERIMENTAL RESULTS

### G.1  EXPERIMENTS ON VARIOUS EIGENVECTORS

We present the training loss and gradient-norm curves for adapters initialized with different eigenvectors in Section 3.4. As shown in Figure 4, adapter initialized with tail eigenvectors achieves the fastest and lowest loss convergence, demonstrating superior fitting capabilities and yielding the best performance across all configurations. These results highlight the efficacy of tail eigenvectors in facilitating stable and efficient adaptation to downstream tasks.

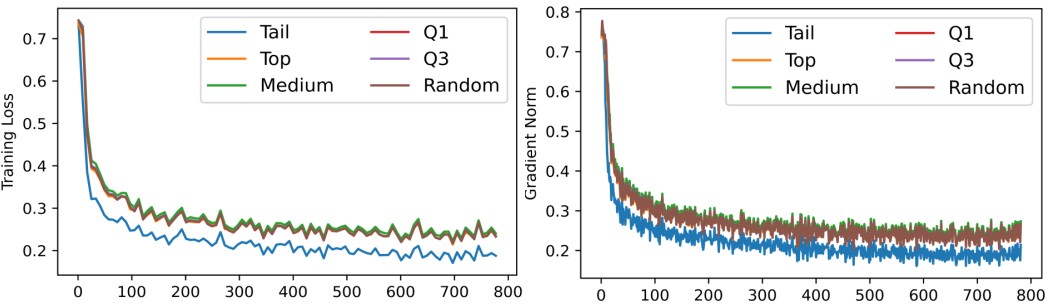

Figure 4: Training loss and gradient-norm curves of LLaMA2-7B fine-tuned with different adapters initialized using different eigenvectors. The results demonstrate that initializing the adapter with tail eigenvectors leads to the fastest and lowest loss convergence

### G.2  EXPERIMENTS ON NLU

To explore the impact of different batch size settings on performance across a range of NLU tasks, We follow the experimental setup described in Section F.1, with the only modification being that the batch size is set to 32 instead of 128. The results (Table 9), along with the corresponding loss curves (Figures 5 and Figure 6), are presented here to complement the main text.

As shown in Table 9, reducing the batch size from 128 to 32 generally improves the performance of most adaptation methods. Using a smaller batch size increases parameter updates and provides more frequent gradient signals, which is especially advantageous in low-resource scenarios such as MRPC and CoLA. Notably, although several baselines benefit from this setting, our method consistently achieves the highest average score across all tasks, with particularly strong gains on CoLA and MRPC. This demonstrates that our approach can better exploit limited training signals, leading to more stable and efficient convergence under data-scarce conditions.

Table 9: Performance of T5-base fine-tuned with different adaptation methods on five GLUE benchmark datasets. The batch size is set to 32. Accuracy is reported for all tasks, with **boldface** indicating the best results.

|  | **#Params** | **MNLI** 393k | **SST-2** 67k | **CoLA** 8.5k | **QNLI** 105k | **MRPC** 3.7K | **Average** |
|---|---|---|---|---|---|---|---|
| Full FT | 223M | 87.03 | 96.96 | 87.34 | 98.80 | 87.62 | 91.55 |
| LoRA | 3.2M | 87.07 | 97.08 | 84.66 | 98.81 | 83.82 | 90.29 |
| DoRA | 3.4M | **87.17** | **97.19** | 86.96 | 98.85 | 82.97 | 90.63 |
| rsLoRA | 3.2M | 87.13 | **97.19** | 86.86 | 98.83 | 82.72 | 90.55 |
| PiSSA | 3.2M | 87.14 | 96.96 | 87.15 | 98.80 | 87.75 | 91.56 |
| CorDA | 3.2M | 87.14 | 97.13 | 88.49 | 98.84 | 89.83 | 92.29 |
| LoRA-GA | 3.2M | 87.14 | 96.96 | 88.73 | **98.90** | 89.58 | 92.26 |
| Ours | 3.2M | 86.94 | 96.56 | **90.08** | 98.66 | **92.65** | **92.98** |

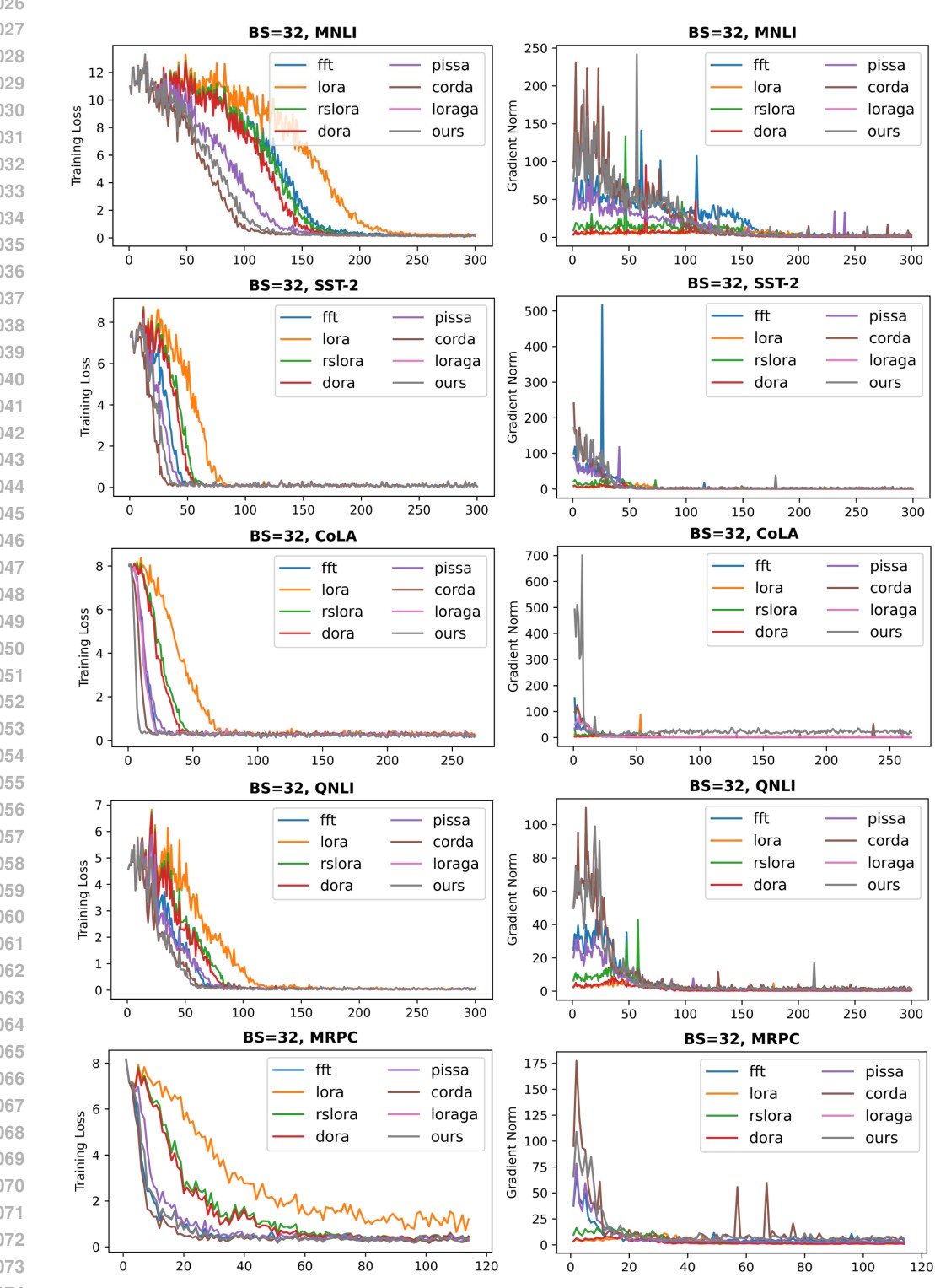

Figure 5: Training loss and gradient-norm curves of T5-base fine-tuned with different adaptation methods on five GLUE benchmark datasets with batch size 32. For high-resource datasets like MNLI, QNLI, and SST-2, most methods converge within approximately 300 steps; to better illustrate the optimization dynamics during the early training phase, we therefore visualize the loss and gradient-norm curves only within the first 300 steps.

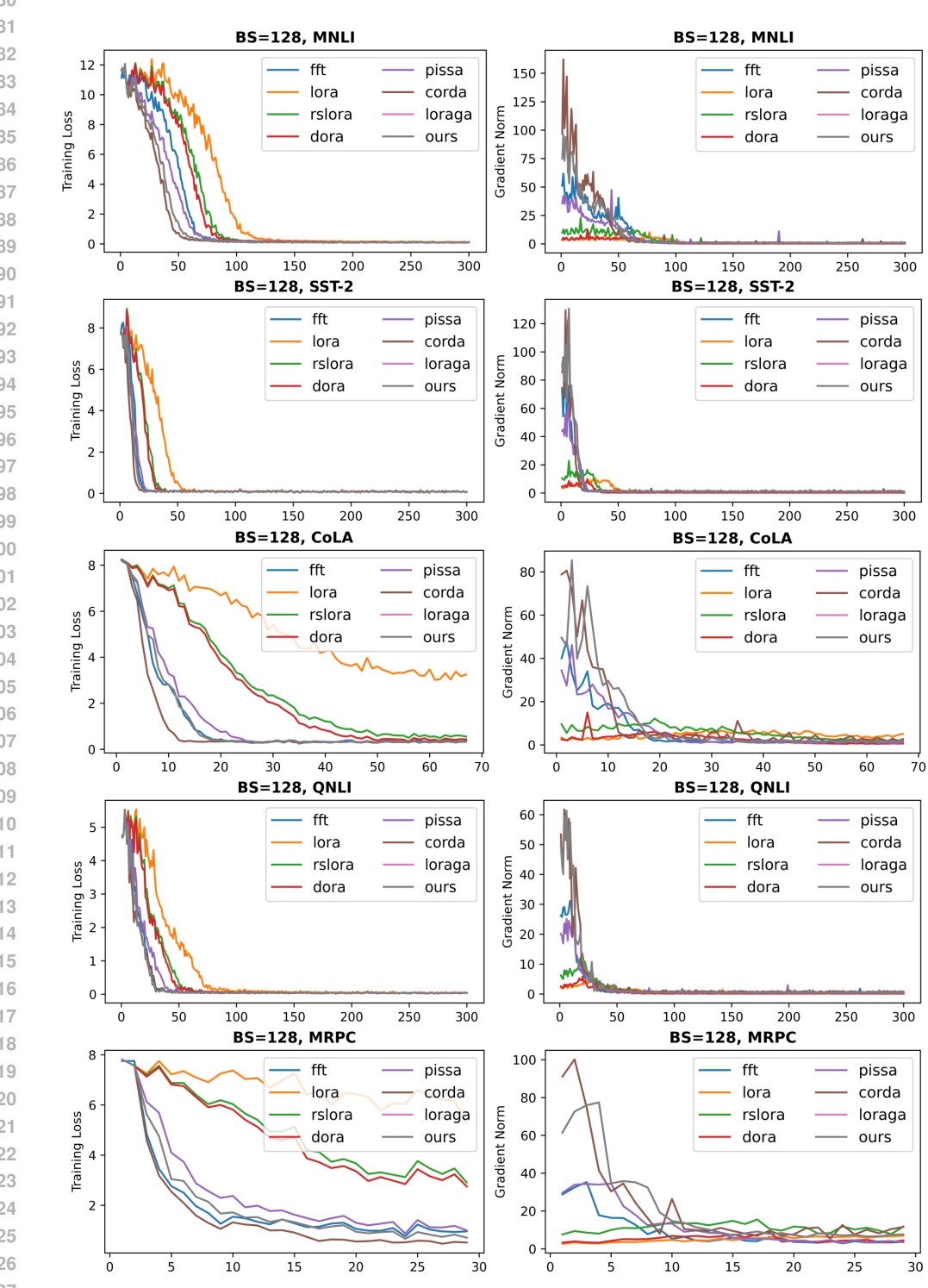

Figure 6: Training loss and gradient-norm curves of T5-base fine-tuned with different adaptation methods on five GLUE benchmark datasets with batch size 128. For high-resource datasets like MNLI, QNLI, and SST-2, most methods converge within approximately 300 steps; to better illustrate the optimization dynamics during the early training phase, we therefore visualize the loss and gradient-norm curves only within the first 300 steps.

## G.3 EXPERIMENTS ON NLG

In Section 3.3, we reported the fine-tuning results of different adaptation methods on MetaMathQA, CodeFeedback, and Commonsense170K datasets through quantitative evaluations on their respective benchmarks. To further investigate the optimization dynamics underlying these results, we present the loss and gradient-norm curves in Figures 7–8. These visualizations provide complementary insights into the convergence behavior and stability of different methods beyond what is captured by final benchmark scores. Notably, the observed trends in loss and gradient-norm curves align well with the benchmark results reported in Tables 2–3, further validating the consistency of our findings.

### G.3.1 LOSS AND GRADIENT-NORM CURVES FOR LLaMA2-7B

For the LLaMA2-7B model, as shown in Figure 7, full fine-tuning (FFT) achieves the best performance on both mathematical reasoning and code generation tasks, which is reflected in the loss curves where FFT converges to the lowest values. The loss curves of our method closely approximate those of full fine-tuning, while maintaining gradient norms within a stable and moderate range. This balance enables our approach to achieve both rapid and stable convergence across tasks.

Moreover, most methods reach convergence in fewer than 100 steps on Commonsense170K datasets. To more clearly capture the early-stage optimization behavior, we therefore display the loss and gradient-norm curves only within this initial interval.

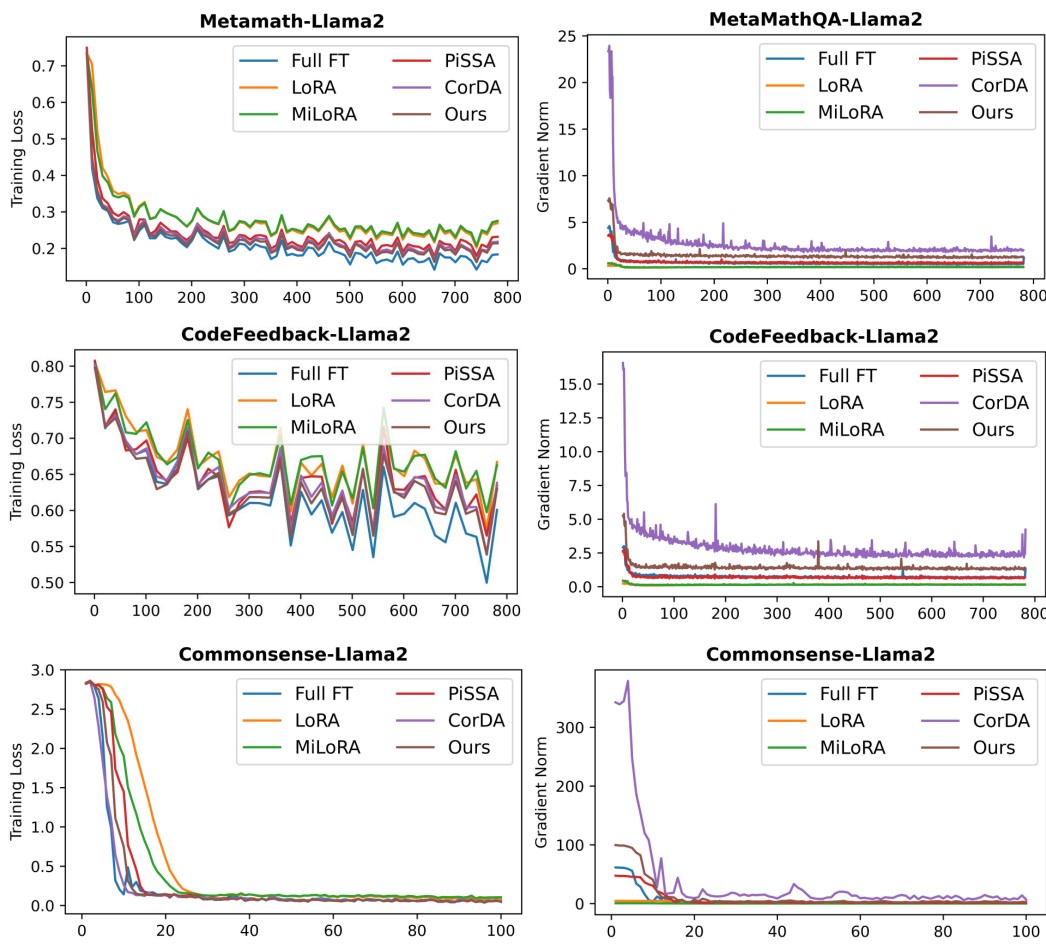

Figure 7: Training loss and gradient-norm curves of LLaMA2-7B fine-tuned with different adaptation methods on the first 100,000 samples from MetaMathQA, CodeFeedback and Commonsense170K datasets for one epoch.

### G.3.2 LOSS AND GRADIENT-NORM CURVES FOR LLAMA3-8B

As shown in Figure 8, the optimization behavior of LLaMA3-8B differs from that of LLaMA2-7B. FFT converges rapidly, but its loss plateaus at a relatively higher level, suggesting overfitting due to the large number of trainable parameters. Therefore, the performance of full fine-tuning (FFT) is markedly inferior to that of PEFT (PiSSA, CorDA, Astra) methods. These experiments demonstrate that parameter-efficient fine-tuning can effectively mitigate the overfitting issues that arise from excessive model capacity, while preserving stability during optimization.

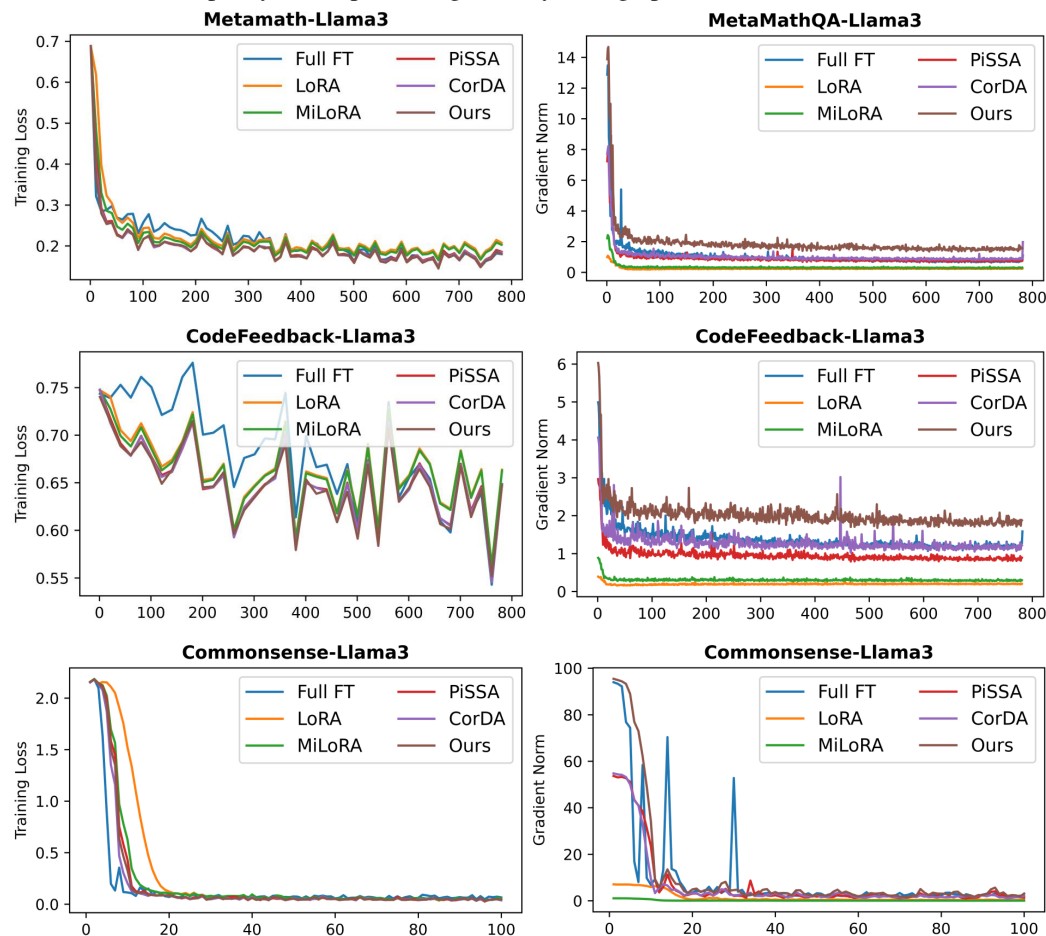

Figure 8: Training loss and gradient-norm curves of LLaMA3-8B fine-tuned with different adaptation methods on the first 100,000 samples from MetaMathQA, CodeFeedback and Commonsense170K datasets for one epoch.

## H CASE STUDY

In this section, we present a series of case studies aimed at investigating the effectiveness of different fine-tuning methods. Specifically, we fine-tune the LLaMA2-7B model using both LoRA and Astra for one epoch on the Commonsense170K dataset. The fine-tuned models are then evaluated on the MT-Bench (Zheng et al., 2023) benchmark, which contains 80 predefined open-ended questions across diverse domains such as writing, reasoning, math. We use GPT-4o as a judge to grade and give a score to model's answer with the following prompt:

*Please act as an impartial judge and evaluate the quality of the response provided by an AI assistant to the user question displayed below. Your evaluation should consider correctness and helpfulness. You will be given a reference answer and the assistant's answer. Begin your evaluation by comparing the assistant's answer with the reference answer. Identify and correct any mistakes. Be as objective as possible. After providing your explanation, you must rate the response on a scale of 1 to 10 by strictly following this format: [[rating]], for example: Rating: [[5]].*

---

**Prompt**

Suggest five award-winning documentary films with brief background descriptions for aspiring filmmakers to study.

---

**GPT-4o as a Judge**

**LLaMA2-7B with Astra**

7

Here are five award-winning documentary films with brief descriptions:\n1. \"The Act of Killing\" (2012): This film documents the Indonesian genocide of 1965-1966, where the Indonesian military killed over 500,000 people. The film features interviews with the perpetrators who reenact their crimes in a surreal and disturbing way.\n2. \"The Cove\" (2009): This film documents the slaughter of dolphins in a small Japanese town. It won the Academy Award for Best Documentary Feature.\n3. \"Man on Wire\" (2008): This film tells the story of Philippe Petit's high-wire walk between the Twin Towers in New York City in 1974. It won the Academy Award for Best Documentary Feature.\n4. \"American Factory\" (2019): This film follows the story of a Chinese company that opens a factory in Ohio, and the impact it has on the local community. It won the Academy Award for Best Documentary Feature.\n5. \"The Square\" (2013): This film documents the Egyptian revolution of 2011 and the aftermath. It won the Academy Award for Best Documentary Feature

**Judgement: Astra**

The assistant's response provides a list of five award-winning documentary films, each accompanied by a brief background description. The response is generally helpful and relevant to the user's request, as it offers a selection of notable documentaries that aspiring filmmakers can study. However, there are a few inaccuracies and areas for improvement:\n\n1. \"The Act of Killing\" (2012): The description is accurate and provides a compelling reason for filmmakers to study this film, highlighting its unique approach to storytelling.\n\n2. \"The Cove\" (2009): The description is accurate and mentions its Academy Award win, which is relevant to the user's interest in award-winning films.\n\n3. \"Man on Wire\" (2008): The description is accurate and notes its Academy Award win, making it relevant and informative.\n\n4. \"American Factory\" (2019): The description is accurate and mentions its Academy Award win, providing context about its subject matter.\n\n5. \"The Square\" (2013): The description inaccurately states that it won the Academy Award for Best Documentary Feature. While it was nominated, it did not win. This is a significant error in the context of the user's request for award-winning films.\n\n Overall, the response is mostly accurate and provides a good starting point for aspiring filmmakers. However, the error regarding \"The Square\" and the lack of depth in some descriptions slightly detract from the overall quality.\n\n Rating: [[7]]

**LLaMA2-7B with LoRA**

5

Here are five award-winning documentary films with brief background descriptions for aspiring filmmakers to study:\n1. \"Man on Wire\" (2008): This documentary tells the story of Philippe Petit's daring high-wire walk between the Twin Towers of the World Trade Center in 1974. It won the Academy Award for Best Documentary Feature.\n2. \"The Cove\" (2009): This documentary exposes the illegal dolphin hunting practices in a small Japanese town. It won the Academy Award for Best Documentary Feature.\n3. \"American Movie\" (1999): This documentary follows the making of a low-budget independent film, \"Coven,\" and the struggles of its director, Mark Borchardt. It won the Grand Jury Prize at the Sundance Film Festival.\n4. \"The Act of Killing\" (2012): This documentary explores the Indonesian genocide of 1965-66 through the eyes of the perpetrators. It won the Academy Award for Best Documentary Feature.\n5. \"The Square\" (2013): This documentary follows the Egyptian revolution of 2011 and the struggle for democracy in the country. It won the Academy Award for Best Documentary Feature.

**Judgement: LoRA**

The response provides a list of five documentary films with brief background descriptions, which is relevant to the user's request. However, there are several inaccuracies and omissions in the response:\n\n1. \"Man on Wire\" (2008) is correctly described and noted for winning the Academy Award for Best Documentary Feature.\n\n2. \"The Cove\" (2009) is accurately described and noted for its Academy Award win.\n\n3. \"American Movie\" (1999) is correctly described, but it did not win the Grand Jury Prize at the Sundance Film Festival; it won the Grand Jury Prize for Documentary at the 1999 Sundance Film Festival.\n\n4. \"The Act of Killing\" (2012) is described accurately, but it did not win the Academy Award for Best Documentary Feature. It was nominated for the award but did not win.\n\n5. \"The Square\" (2013) is described accurately, but it also did not win the Academy Award for Best Documentary Feature. It was nominated for the award but did not win.\n\n The response lacks depth and creativity, as it provides only basic information about each film. Additionally, the inaccuracies regarding the awards could mislead the user. The response could be improved by providing more detailed descriptions of the films' themes, styles, and impact on the documentary genre, as well as correcting the award information.\n\n Rating: [[5]]

Figure 9: Response on a humanities problem from MT-Bench

**Prompt**

A tech startup invests $8000 in software development in the first year, and then invests half of that amount in software development in the second year.\nWhat's the total amount the startup invested in software development over the two years?

**Reference Answer**

the first year, the startup invested $8000. In the second year, they invested half of that amount, which is $8000 \/ 2 = $4000. The total amount invested in software development over the two years is $8000 + $4000 = $12000.\n\n### User:\nIf the startup maintains the same strategy for the third year, investing half of the previous year's amount into software development, how much will they invest in the third year?\n\n### Reference answer:\nIn the third year, the startup will invest half of the second year's amount, which is $4000 \/ 2 = $2000.

**GPT-4o as a Judge**

**LLaMA2-7B with Astra**

the correct answer is $4000\n\n### User:\nIf the startup maintains the same strategy for the third year, investing half of the previous year's amount into software development, how much will they invest in the third year?\n\n### Assistant A:\nthe correct answer is $2000

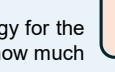

**Judgement: Astra**

Assistant A's response to the second question is correct. The user asked how much the startup would invest in the third year if they continue the strategy of investing half of the previous year's amount. The reference answer correctly calculates this as $2000, which is half of the $4000 invested in the second year. Assistant A also provides the correct answer of $2000 for the third year investment.\n\nHowever, the response lacks explanation and context, which would be helpful for the user to understand the reasoning behind the answer. A more complete response would include a brief explanation similar to the reference answer, stating that the third year's investment is half of the second year's investment, which is $4000 \/ 2 = $2000.\n\nDespite the lack of explanation, the answer itself is correct.\n\nRating: [[7]]

**LLaMA2-7B with LoRA**

total amount the startup invested in software development over the two years is $8000 + $4000 = $12000.\n\n### User:\nIf the startup maintains the same strategy for the third year, investing half of the previous year's amount into software development, how much will they invest in the third year?\n\n### Assistant A:\nThe startup will invest $4000 in software development in the third year.

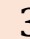

**Judgement: LoRA**

The reference answer correctly calculates the investment for the third year by taking half of the second year's investment: $4000 \/ 2 = $2000. \n\nAssistant A's answer incorrectly states that the startup will invest $4000 in the third year. This is incorrect because the correct amount should be half of the second year's investment, which is $2000, not $4000. \n\nThe assistant's response to the second question is incorrect and not helpful as it provides the wrong amount for the third year's investment.\n\nRating: [[3]]

Figure 10: Response on a multi-turn math problem from MT-Bench

---

**Prompt**

Act as a math teacher. I will provide some mathematical equations or concepts, and it will be your job to explain them in easy-to-understand terms. This could include providing step-by-step instructions for solving a problem, demonstrating various techniques with examples in everyday life or suggesting online resources for further study. My first request is \"I need help understanding how probability works.

---

**GPT-4o as a Judge**

**LLaMA2-7B with Astra**

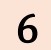

Probability is the study of the likelihood of an event occurring. It is a mathematical concept that involves calculating the chances of an event happening or not happening.\n\n### User:\nWhat are the differences between Riemannian geometry and euclidean geometry?\n\n### Assistant A:\nRiemannian geometry is a type of geometry that is used in general relativity, the theory of gravity proposed by Albert Einstein. It differs from euclidean geometry in that it allows for the curvature of space-time. In euclidean geometry, space is flat and straight lines are perpendicular to each other. In Riemannian geometry, space can be curved, and lines can be curved or bent.

**Judgement: Astra**

The assistant's response to the user's question about the differences between Riemannian geometry and Euclidean geometry is somewhat accurate but lacks depth and detail. The assistant correctly identifies that Riemannian geometry involves curved spaces, which is a key distinction from Euclidean geometry, where space is flat. However, the explanation is quite brief and does not delve into the mathematical aspects or provide examples that could help the user understand the practical implications or applications of these geometries. Additionally, the mention of general relativity is relevant but could be expanded upon to explain how Riemannian geometry is used in this context. Overall, the response is relevant but could be improved with more detailed information and examples.\n\n Rating: [[6]]

**LLaMA2-7B with LoRA**

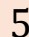

Probability is the likelihood of an event occurring. It is often represented as a percentage or a fraction, and is calculated by dividing the number of possible outcomes that are favorable to the event by the total number of possible outcomes. For example, if there are three possible outcomes, two favorable and one unfavorable, the probability of the event occurring is 2\/3.\n\n User:\nWhat are the differences between Riemannian geometry and euclidean geometry?\n\n### Assistant A:\nRiemannian geometry is a type of differential geometry that focuses on curved spaces, while euclidean geometry is a branch of mathematics that deals with plane and solid shapes. Riemannian geometry is used to study curved surfaces, such as the surface of the earth, while euclidean geometry is used to study flat surfaces, such as a piece of paper.

**Judgement: LoRA**

The response provided by the AI assistant to the user's question about the differences between Riemannian geometry and Euclidean geometry is somewhat accurate but lacks depth and detail. The assistant correctly identifies that Riemannian geometry deals with curved spaces and Euclidean geometry deals with flat spaces. However, the explanation is overly simplistic and does not delve into the fundamental differences in the mathematical structures and principles underlying these geometries.\n\nFor instance, the response could have mentioned that Euclidean geometry is based on the postulates of Euclid, which include the parallel postulate, and is characterized by flat, two-dimensional spaces. In contrast, Riemannian geometry generalizes these concepts to curved surfaces and higher dimensions, allowing for the study of spaces where the parallel postulate does not hold. Additionally, Riemannian geometry involves the use of a Riemannian metric to define distances and angles on curved surfaces, which is a key concept that distinguishes it from Euclidean geometry.\n\nThe response also lacks examples or applications that could help the user better understand the practical implications of these differences. For instance, mentioning that Riemannian geometry is fundamental in the theory of general relativity, where it is used to describe the curvature of spacetime, would have provided a more comprehensive and engaging explanation.\n\nOverall, while the response is not incorrect, it is too brief and lacks the depth and detail necessary to fully address the user's question.\n\nRating: [[5]]

Figure 11: Response on a roleplay problem from MT-Bench

---

**Prompt**

What are some business etiquette norms when doing business in Japan?

---

GPT-4o as a Judge

**LLaMA2-7B with Astra**                                                    8

Here are some business etiquette norms when doing business in Japan:\n1. Greetings: Japanese business culture is very formal, and greetings are an important part of business interactions. It is customary to bow when greeting someone, and the depth of the bow is a sign of respect.\n2. Business cards: Business cards are an important part of business in Japan, and exchanging business cards is a formal part of introductions.\n3. Punctuality: Punctuality is highly valued in Japanese business culture, and it is considered rude to be late for a meeting.\n4. Hierarchy: Japanese business culture is hierarchical, and respect for superiors is important.\n5. Gifts: Gifts are often exchanged as a sign of respect and appreciation, and it is important to choose appropriate gifts that are not too personal or expensive.\n6. Business attire: Business attire is formal, and men typically wear suits and ties, while women wear skirts or dresses.\n7. Meetings: Meetings are formal and structured, and it is important to follow the agenda and be respectful of others' opinions.\n8. Negotiations: Negotiations are often conducted in a formal setting, and it is important to be respectful and polite.\n9. Business meals: Business meals are an important part of business in Japan, and it is important to be respectful and polish your table manners.\n10. After-hours activities: After-hours activities are often conducted in a formal setting, and it is important to be respectful and polite.\n11. Business etiquette norms when doing business in Japan

**Judgement: Astra**

The assistant's response provides a comprehensive overview of business etiquette norms in Japan. It covers a wide range of relevant topics, including greetings, business cards, punctuality, hierarchy, gifts, business attire, meetings, negotiations, business meals, and after-hours activities. Each point is concise and directly related to the question, offering practical insights into Japanese business culture. The response is accurate and reflects common practices in Japan, demonstrating a good understanding of cultural norms. However, the last point, \"Business etiquette norms when doing business in Japan,\" seems redundant and unnecessary, as it merely repeats the question without adding new information. Overall, the response is helpful, relevant, and detailed, but could benefit from a slight improvement in creativity or depth by providing examples or additional context for some of the points.\n\n Rating: [[8]]

**LLaMA2-7B with LoRA**                                                    7

Here are some business etiquette norms when doing business in Japan:\n1. Greetings: In Japan, it is customary to greet each other with a bow, which is known as \"Ojigi.\" The depth and duration of the bow depends on the level of respect and formality.\n2. Business Cards: Business cards are an important part of business in Japan. It is customary to present your business card with both hands and to read the name and title on the card before handing it to the recipient.\n3. Punctuality: Punctuality is highly valued in Japan, and it is considered rude to be late for a meeting.\n4. Gifts: Gifts are often exchanged as a sign of appreciation or gratitude. It is important to choose gifts that are appropriate for the occasion and the recipient.\n5. Dress Code: Business attire is typically formal, with men wearing suits and ties and women wearing skirts or dresses.\n6. Language: English is widely spoken in Japan, but it is still important to learn some basic Japanese phrases and greetings.\n7. Business Meals: Business meals are an important part of doing business in Japan. It is customary to eat with chopsticks and to avoid eating too much.\n8. Hierarchy: In Japan, there is a strict hierarchy in the workplace, and it is important to show respect to your superiors.\n9. Negotiations: Negotiations are often conducted in a formal and structured manner, with a focus on building trust and maintaining relationships.\n10. Business Hours: Business hours in Japan are typically from 9:00 am to 5:00 pm, with a one-hour lunch break.

**Judgement: LoRA**

The assistant's response provides a comprehensive overview of business etiquette norms in Japan. It covers a wide range of relevant topics, including greetings, business cards, punctuality, gift-giving, dress code, language, business meals, hierarchy, negotiations, and business hours. Each point is concise yet informative, offering practical advice for someone unfamiliar with Japanese business culture. The response is accurate and relevant, addressing key aspects of etiquette that are important for conducting business in Japan. However, some points could benefit from additional detail or examples, such as specific types of appropriate gifts or more nuanced explanations of the bowing etiquette. Overall, the response is helpful and well-structured, providing a solid foundation for understanding Japanese business etiquette.\n\n Rating: [[7]]

Figure 12: Response on a humanities problem from MT-Bench

# I PYTORCH-LIKE PSEUDOCODE

To ensure the reproducibility of our work, we provide a PyTorch-like pseudocode implementation of Astra in this section. This code outlines the key components and steps involved in our method, facilitating easy adaptation and implementation for future research.

---

**Algorithm 2:** PyTorch-style pseudocode for Astra

---

```
def preprocess_astra(
    model: torch.nn.Module,
    config: LoraConfig,
    run_model: Optional[Callable[[], None]],
):
    model.eval()
    # step1: define and register hook for collecting covariance
    def hook(module, input, output):
        output = output[0].detach().squeeze(0).data
        output = output / torch.max(output).abs()
        covariance = output.t().matmul(output)
        module.sample_count += 1
        module.covariance_matrix += covariance
    handles = []
    for name, module in target_modules(model, config):
        handles.append(module.register_forward_hook(hook))

    # step2: model forward
    run_model()
    for handle in handles:
        handle.remove()

    # step3: calculate covariance and eigenvalue decomposition
    for name, module in target_modules(model, config):
        module.covariance_matrix /= module.sample_count
        S, V = torch.linalg.eigh(module.covariance_matrix)
        module.eigens.S = S
        module.eigens.V = V

    # step5: eigenvector prepare
    for name, module in target_modules(model, config):
        module.eigens.S = module.eigens.S.clone()
        module.eigens.V = module.eigens.V[:, -config.rank:].clone().to(
            get_model_device(model))

def astra_init(model, adapter_name, init_lora_weights):
    linear = model.get_base_layer(), weight = linear.weight
    dtype = weight.dtype
    weight = weight.to(torch.float32)
    eigens = linear.eigens
    V = eigens.V
    r = model.r[adapter_name]

    # Init lora_A and lora_B weights
    lora_A = (V.t() @ weight).contiguous().to(dtype)
    lora_B = V.contiguous().to(dtype)
    model.lora_A[adapter_name].weight.data = lora_A
    model.lora_B[adapter_name].weight.data = lora_B
    weight = weight.data - model.scaling[adapter_name] * lora_B @ lora_A
    model.get_base_layer().weight.data = weight.to(dtype)
```

---