# OpenReview forum: "Astra: Activation-Space Tail Eigenvector Low-Rank Adaptation of Large Language Models"
_ICLR.cc/2026/Conference — ICLR 2026 Conference Withdrawn Submission_

### Official Review · Reviewer_BMu6 · 2025-10-29

**Soundness:** 1
**Presentation:** 2
**Contribution:** 1
**Rating:** 2
**Confidence:** 4

**Summary:**

This paper introduces ASTRA, a representation-based fine-tuning method that projects pretrained model outputs onto the tail subspace of the activation covariance matrix. Instead of updating model parameters, it applies low-rank projections in activation space to minimize perturbation energy. The approach is theoretically motivated and compared to existing methods like ReFT and LoRA.

**Strengths:**

- The use of the hidden representation space is an interesting direction.

- The paper is clearly written and easy to follow.

- A variety of experiments and analyses are conducted.

**Weaknesses:**

- W1. The proposed method uses tail eigenvectors to minimize perturbation energy. However, recent papers, including LoRA-GA [1], theoretically suggest that initialization should be used to approximate the overall fine-tuning gradient. In this regard, Astra's initialization method appears to limited capacity for early-stage learning. It would be helpful to provide a theoretical justification for why using tail eigenvectors is beneficial.

- W2. The approach using covariance is similar to CorDA [2], but the necessity of initializing based on data covariance is not clearly explained. Spectral-based initialization methods like PiSSA [3] and MiLoRA [4] also decompose the weight matrix into orthogonal main and tail components and aim to preserve information via minor components. The motivation for the proposed method may therefore seem insufficiently distinctive.

- W3. The mathematical comparison to ReFT [5] is not entirely convincing. Authors said that the equivalence holds when $R$ equals $W$ and $b=0$. However, in ReFT, $R$ is an orthogonal matrix and $W$ is a learnable projection weight, so the two are rarely the same. Assuming $R = W$ and $b=0$ results in $Y=Y$ for line 441 equation, and treating the resulting decomposition as equivalent to Astra may oversimplify the interpretation. If such logic were followed, other spectral initialization methods like PiSSA and MiLoRA could also be considered equivalent to ReFT, which is not generally accepted.

- W4. Theorem 2.1 appears to reiterate well-known properties. The orthogonality of bases in EVD or SVD is a basic mathematical fact, and presenting it as a formal theorem might be excessive.

- W5. For Theorem 2.2, it would strengthen the contribution to explain how the stated condition supports effective adaptation, such as faster convergence or stable fine-tuning.

> [1] Wang, Shaowen, Linxi Yu, and Jian Li. "Lora-ga: Low-rank adaptation with gradient approximation." *Advances in Neural Information Processing Systems* 37 (2024): 54905-54931.
>
> [2] Yang, Yibo, et al. "CorDA: Context-Oriented Decomposition Adaptation of Large Language Models." *CoRR* (2024).
>
> [3] Meng, Fanxu, Zhaohui Wang, and Muhan Zhang. "Pissa: Principal singular values and singular vectors adaptation of large language models." *Advances in Neural Information Processing Systems* 37 (2024): 121038-121072.
>
> [4] Wang, Hanqing, et al. "MiLoRA: Harnessing Minor Singular Components for Parameter-Efficient LLM Finetuning." *Proceedings of the 2025 Conference of the Nations of the Americas Chapter of the Association for Computational Linguistics: Human Language Technologies (Volume 1: Long Papers)*. 2025.
>
>[5] Wu, Zhengxuan, et al. "Reft: Representation finetuning for language models." Advances in Neural Information Processing Systems 37 (2024): 63908-63962.

**Questions:**

- Q1.  While Astra is conceptually different from PCA, the idea of projecting based on the covariance structure of activations naturally evokes the PCA framework with minor component direction. However, while PCA projects onto the basis of hidden representations, Astra applies the basis to the weight space, making the geometric interpretation less intuitive.

- Q2. The code link appears to be broken or inaccessible. Please verify that the repository functions correctly.

---

### Official Review · Reviewer_muFY · 2025-10-30

**Soundness:** 2
**Presentation:** 4
**Contribution:** 3
**Rating:** 6
**Confidence:** 4

**Summary:**

In this paper, the authors first revealed two challenges underlying LoRA parameter initialization: (1) the output activations of LLM exhibit low-rank structure, where the major components are captured in a low-dimensional subspace. This principle low-rank subspace is progressively formed and optimized during pretraining to capture rich semantic information. However, further updates within this subspace during fine-tuning yield diminishing return, potentially disturbing the learned representations and causing unstable convergence. (2) Meanwhile, the dimensions corresponding to tail eigenvalues remain under-utilized. To address these two challenges, the authors proposed Astivation-Space Tail-Eigenvector Low-Rank Adaptation (Astra), which leverages the under-utilized "tail eigenspace" of model output activations to initialize LoRA adapters. The authors claimed that Astra possesses two advantages: (1) Orthogonality to task-relevant pretrained semantic structure: By initializing LoRA adapters in directions orthogonal to the principal activation subspace, Astra minimizes interference with the model’s native task competence, ensuring stability and semantic consistency during fine-tuning, （2）Energy-minimizing initialization: Among all possible low-rank update directions, Astra selects those that minimize perturbation energy in the original task-relevant output space.

**Strengths:**

1. The paper is well-written and easy to follow;

2. The motivation of this paper is reasonable;

3. The proposed Astra got mathematical support.

4. The proposed method can improve the convergence speed.

5. The proposed method is robust to calibration data.

**Weaknesses:**

1. The experiments are only conducted on Llama-family models, the applicability of the Astra method to other LLMs, like Qwen and Mistral, is questionable.

2. Compared to the original LoRA, the proposed method can improve the convergence speed. While there exists much work that focuses on improving the initialization of LoRA to boost the convergence speed, the authors should involve more STOA baselines in the convergence speed analysis to support the claimed advantage of Astra.

3. The authors claimed that the proposed Astra can enhance LoRA's training stability, while they did not provide experimental analysis to support this claim.

**Questions:**

Please see the Weaknesses.

---

### Official Review · Reviewer_Qnbx · 2025-11-03

**Soundness:** 2
**Presentation:** 2
**Contribution:** 2
**Rating:** 2
**Confidence:** 4

**Summary:**

This paper proposes  simple initialization scheme for LoRA consisting in estimating the low dimensional subspace corresponding to the smallest eigenvectors of the output covariance of a layer to be fine-tuned with LoRA. The adapter is initialized to the the restriction of the weight matrix to this low rank subspace, while the frozen part is set to the residual, which corresponds to conserving in the frozen weight the information that is most relevant to the output's principal directions.

The proposed method is mainly supported by an extensive experimental section. A variation on the Eckart-Young-Mirsky  theorem is provided to theoretically motivate the approach.

The experimental section shows that the proposed initialization method outperform other initialization schemes and LoRA variants, sometimes by a large margin.

**Strengths:**

- The idea presented in this submission is quite simple and intuitive.

- Experiments show that the proposed method is quite effective despite its simplicity.

**Weaknesses:**

- The abstract and introduction are in some sense misleading: it is stated in several places that the method "constrains updates to the subspace". As far as I understand this is not true, only the initialization is such that the adapter lies in the subspace, but nothing is done during fine tuning to enforce the adapter to remain "constrained" to this subspace.

- The theoretical analysis is quite limited:
  - Theorem 2.2 is in essence a straightforward linear algebra result that does not connect very well with the end goal of the proposed method. I would suggest the authors to
   1) motivate and explain better why "minimizing output perturbation energy" is a desirable property for fine-tuning  with LoRA.
   2) look at the literature on low rank regression (a.k.a. reduce rank regression). I think there are some deep connections between the proposed approach and this old statistical technique that could better anchor the contribution in principled theory.
  - Theorem 2.1 is itself a very informal remark that, in my opinion, should not be stated as a theorem since the statement is very informal and can have multiple interpretations (e.g. Y_main, Y_tail has not been properly defined).

- There is a slight lack of mathematical rigor in some place. In particular, I suggest the authors to introduce their notations (e.g. M[:,-r:] denotes the last r column of a matrix..., why is x lower case and Y upper case in Eq 3-5? ).

- The accuracies/errors reported in Table 1 and 2 are quite different from the ones reported in the two main competitors: PiSSa and MiLoRa, making it a bit difficult to isolate the superiority of the method to the method itself, or the experimental setup that seems to differ from this 2 other work (even though it is stated that the experimental setup of MiLoRa is used).

**Questions:**

- Why are the value reported in Tables 1 and 2 so different from the ones reported in PiSSa and MiLoRa? Ideally, we would like the results to be relatively similar the ones in PiSSa, or at least clearly understand why they are so different.

---

### Official Review · Reviewer_YGy4 · 2025-11-08

**Soundness:** 3
**Presentation:** 3
**Contribution:** 2
**Rating:** 4
**Confidence:** 3

**Summary:**

This paper proposes Astra, a PEFT method that initializes LoRA adapters using tail eigenvectors of output activation covariance matrices. The authors provide theoretical justification showing that tail subspaces minimize output perturbation energy and remain orthogonal to pretrained semantic structure. Extensive experiments across 16 benchmarks demonstrate performance improvements over existing LoRA variants.

**Strengths:**

This paper proposes Astra, a novel PEFT method that initializes LoRA adapters using tail eigenvectors of output activation covariance matrices.
1. The paper maintains a logical flow with good academic structure, clear figure presentations, and detailed proofs, etc.
2. Extensive experiments show improvements across 16 diverse benchmarks.

**Weaknesses:**

1. The paper claims to "minimize output perturbation" (Theorem 2.2) as a design principle, yet fine-tuning inherently requires changing model outputs to adapt to new tasks. This paradox isnt addressed: why should minimizing perturbations lead to better task adaptation? The causal link between "small initial perturbation" and "superior final performance" remains unexplained.

2. The paper omits Astra's preprocessing overhead and and specify when this overhead is justified given the often marginal performance gains (e.g., +0.12% on MNLI). eg. 64 forward passes plus O(d³) eigendecomposition for 200+ modules (over 10T operations for LLaMA2-7B), right?

3. Despite claiming that tail subspace optimization "avoids interfering with pretrained semantic structure", no experiment proves that. All experiments follow single-task evaluation (train on A, test on A) with no catastrophic forgetting related analysis, or experiments.


4. The paper contains no limitations section and never discusses scenarios where Astra underperforms.

5. Why does performance vary so dramatically across tasks?
For example, in Table 1, Astra improves MRPC , yet on SST-2 it decreases from 96.62% to 96.45% (-0.17%). What task characteristics determine whether Astra is suitable? Under what conditions does Astra fail to work effectively?

**Questions:**

See the questions in each weakness point above.

---

### Note · Authors · 2026-01-06

I have read and agree with the venue's withdrawal policy on behalf of myself and my co-authors.